# Fluorinated polyamidoamine dendrimer-mediated miR-23b delivery for the treatment of experimental rheumatoid arthritis in rats

Haobo Han [1], Jiakai Xing[1], Wenqi Chen[1], Jiaxin Jia[1] & Quanshun Li [1] ✉

In rheumatoid arthritis (RA), insufficient apoptosis of macrophages and excessive generation of pro-inflammatory cytokines are intimately connected, accelerating the development of disease. Here, a fluorinated polyamidoamine dendrimer (FP) is used to deliver miR-23b to reduce inflammation by triggering the apoptosis of as well as inhibiting the inflammatory response in macrophages. Following the intravenous injection of FP/miR-23b nanoparticles in experimental RA models, the nanoparticles show therapeutic efficacy with inhibition of inflammatory response, reduced bone and cartilage erosion, suppression of synoviocyte infiltration and the recovery of mobility. Moreover, the nanoparticles accumulate in the inflamed joint and are non-specifically captured by synoviocytes, leading to the restoration of miR-23b expression in the synovium. The miR-23b nanoparticles target *Tab2*, *Tab3* and *Ikka* to regulate the activation of NF-κB pathway in the hyperplastic synovium, thereby promoting anti-inflammatory and anti-proliferative responses. Additionally, the intravenous administration of FP/miR-23b nanoparticles do not induce obvious systemic toxicity. Overall, our work demonstrates that the combination of apoptosis induction and inflammatory inhibition could be a promising approach in the treatment of RA and possibly other autoimmune diseases.

Rheumatoid arthritis (RA) is one of the most common chronic inflammatory autoimmune diseases, which affects 0.5–1.0% of the population worldwide[1]. Clinically, the hyperplasia and infiltration of macrophages, lymphocytes and fibroblast-like synoviocytes (FLSs) has been demonstrated to be the main pathophysiological characteristics in RA progression, leading to irreversible bone destruction, cartilage erosion and even high risk of disability[2]. To date, the precise mechanism contributing to the pathogenesis of RA is still unclear and imperative to be elucidated, and the therapeutic drugs primarily rely on the inhibition of inflammatory response[3]. Although significant progress has been achieved in the RA treatment especially at the early stage of RA, the currently approved approaches still suffer certain limitations[4,5]. For instance, the use of biologics to inhibit the

interleukin-1 (IL-1) and tumor necrosis factor (TNF) originated from the synovial macrophages, has been considered to be an effective amelioration on the symptoms of RA[6–9], but only 40% of patients could respond to the therapy against IL-1 and TNF. Meanwhile, the disease will recur once the treatment stops, indicating that this type of therapy is suppressive, not curative[10,11]. Not surprisingly, the sustained inflammation of RA is coordinated by a network composed of extensive cytokines, and therefore the inhibition of one or a few may not be efficient to prevent the development of RA[12,13]. In addition, the consecutively secreted molecules can innate the activation of synovial macrophages, resulting in their proliferation and accumulation in the inflamed synovium, whereas the abundant macrophages, in turn, produce widespread pro-inflammatory cytokines, developing the

[1]Key Laboratory for Molecular Enzymology and Engineering of Ministry of Education, School of Life Sciences, Jilin University, 130012 Changchun, China.
✉e-mail: quanshun@jlu.edu.cn

vicious circle to accelerate the disease progression[14]. Moreover, after being activated by the sustained pro-inflammatory cytokines, the primed macrophages eventually possess resistance to cell death signals[15]. Therefore, not only the inflammatory inhibition but also the apoptosis induction of activated macrophages seems to be necessary for diminishing the local chronic inflammation[13,16].

MicroRNAs (miRNAs) are a type of noncoding oligonucleotides with the ability to modulate the gene expression at the post-transcriptional level[17]. They have been found to be highly associated with inflammatory autoimmune diseases including systemic lupus erythematosus (SLE), multiple sclerosis (MS) and RA[18]. Recently, miR-23b was identified to be a key suppressor in the inflamed joints, which could inhibit the pathogenesis and development of RA[19,20]. Through its influence on the nuclear factor (NF)-κB signaling pathway (a crucial driver of inflammation)[21], miR-23b achieves the balance in the resistant cells (for instance, FLSs) lining on the inflamed synovium. Meanwhile, miR-23b can further decrease the excessive secretion of pro-inflammatory cytokines like TNF, IL6, and IL1B, and diminish the autoimmune inflammation[22]. Thus, the miR-23b-mediated regulation of NF-κB signaling pathway has been considered as an important mechanism to modulate the magnitude and duration of inflammatory response. Except for this, previous reports have shown that miR-23b could inhibit tumor growth in many malignancies both in vitro and in vivo[23–25]. Unlike the research of oncology, limited research has been conducted to confirm the function of miR-23b in triggering a similar response in RA, such as the induction of apoptosis. Therefore, we are wondering whether the delivery of miR-23b will achieve inflammatory inhibition through regulating the downstream signaling of cytokines' receptors, as well as simultaneously activate the apoptotic effect in inflamed synovium.

Since the unprotected miRNA molecules are easily eliminated by renal clearance system as well as nuclease in the bloodstream, a safe and effective carrier is highly required to facilitate the systemic delivery of miRNA[26]. In contrast to viral carriers, nonviral delivery systems developed rapidly in recent years with dominant properties such as efficient DNA package, ease of production and relatively lower immunogenicity[27]. In addition, nonviral carriers can selectively accumulate in arthritic joints through a process known as ELVIS (Extravasation through Leaky Vasculature and subsequent Inflammatory cell-mediated Sequestration), which is analogous to EPR (enhanced permeability and retention) effect in solid tumors[28]. Among the nonviral carriers, polyamidoamine (PAMAM) dendrimer has been explored as an efficient gene carrier due to its unique properties such as excellent solubility, high charge density, low density, and convenient modification[29,30]. Besides, PAMAM is hypothesized to facilitate the endosome escape through the proton-sponge effect, which protects nucleic acids from enzymatic degradation in the lysosome. However, carboxyl groups formed at the amide bond cleavage of PAMAM decrease the positive charge density of dendrimer and may influence the transfection process[30]. Thus, to further improve the transfection efficiency and safety of PAMAM dendrimer, surface-engineered technologies have been widely adopted to construct its derivatives, such as the modification using polyethylene glycol[31], amino acids[32], carboxylic acid[33], folate acid[34], chondroitin sulfate[35] and nucleobase analog[36,37]. Recently, a fluorination strategy has been developed to dramatically improve the transfection efficacy of PAMAM and reduce its cytotoxicity[38], and the fluorinated PAMAM-mediated gene or protein transfection possessed superior capacity in tumor therapy[39–43]. Moreover, it is noteworthy that the PAMAM dendrimers and their derivatives were not only effective as the gene carrier, but also exhibited intrinsic and unexpected anti-inflammatory properties in experimental arthritis models through the inhibition of cyclooxygenase (COX) activity, which endowed them with the enduring appeal in the RA treatment[44].

In this study, we develop a fluorinated PAMAM (termed as FP) to deliver miR-23b for the treatment of experimental arthritis models. The FP/miR-23b nanoparticles effectively induce cell apoptosis and anti-inflammatory response in macrophages by the inhibition of the NF-κB signaling pathway. In addition, the intravenous injection of the FP/miR-23b nanoparticles demonstrate passive accumulation in the arthritic joints and further alleviate the symptoms of disease. Moreover, the FP/miR-23b nanoparticles show negligible toxicity to the major organs. In summary, treatment with nanoparticles harboring miR-23b inhibits the proliferation of activated macrophages and chronic inflammation, which shows promise in miRNA-based therapy for RA treatment.

## Results

### Synthesis and characterization of fluorinated PAMAM

In recent years, dendrimer PAMAM has been widely utilized in gene delivery, but its transfection efficiency was still limited especially in synoviocytes[45]. To address this issue, a facile fluorination method was adopted to improve its transfection efficiency, in which the amine groups of PAMAM were modified with heptafluorobutyric anhydride to construct a fluorinated derivative, namely FP (Fig. 1a). According to the ¹⁹F NMR spectra (Supplementary Fig. 1), the characteristic peaks of heptafluorobutyric anhydride were observed in FP, and the peak (a) in heptafluorobutyric anhydride significantly shifted to the peak (a') in FP after the conjugation, indicating that PAMAM was successfully modified with heptafluorobutyric acid. Next, to measure the fluorination degree of FP, the number of perfluoro acid on the surface of each FP dendrimer was measured by ninhydrin assay[38]. As shown in Supplementary Fig. 2, the amine groups available on the surface of FP were calculated to be 98.75, suggesting that approximately 29 of the primary amine groups were modified by heptafluorobutyric acid. Moreover, MALDI-TOF MS analysis was conducted to confirm the molecular weight of unmodified PAMAM and FP. As shown in Supplementary Fig. 3, a general molecular weight distribution was well observed, where the molecular weight values of PAMAM and FP were calculated to be 28859.7 and 35017.4, respectively. Therefore, the theoretical number of heptafluorobutyric acid modified on the surface of FP was measured to be ~30, which was consistent with the results obtained by ninhydrin assay. To verify the effect of chemical modification of PAMAM on the performance of DNA binding and condensation, the heptafluorobutyric acids were replaced by butyric acids to obtain the butyric acid-modified PAMAM (termed as HP) using an identical synthesis route. As shown in Supplementary Fig. 4, HP achieved an average number of 28 butyric acids conjugated to each PAMAM molecule, whose conjugation extent was close to FP. The derivatives FP and HP were incubated with miR-23b plasmids at different N/P ratios, respectively, and the formed nanocomplexes were evaluated using gel retardation assay (Fig. 1b and Supplementary Fig. 5). The carriers HP and FP could effectively condense the miR-23b plasmid to form nanoparticles at N/P ratios of >2.0, as the reduced fluorescence intensity of ethidium bromide could be observed when N/P ratios were higher than 2.0. In contrast, the PAMAM/miR-23b nanoparticles initiated the complexation at the N/P ratio of 0.5 and realized the complete retardation at the N/P ratio of <2.0. These results suggested that the DNA binding ability of surface-engineered PAMAM derivates was slightly reduced in comparison to the unmodified counterparts. Moreover, the influence of PAMAM derivates on the miR-23b complexation was also tested by zeta potential measurement. As shown in Supplementary Fig. 6a, both FP/miR-23b and HP/miR-23b nanoparticles shared similar zeta potential values, which were lower than PAMAM/miR-23b nanoparticles at the identical N/P ratios. The decreased zeta potential values of FP/miR-23b and HP/miR-23b nanoparticles were caused by the loss of positively charged primary amine groups on PAMAM after the chemical modification, further leading to the weakened binding and condensation ability of plasmids.

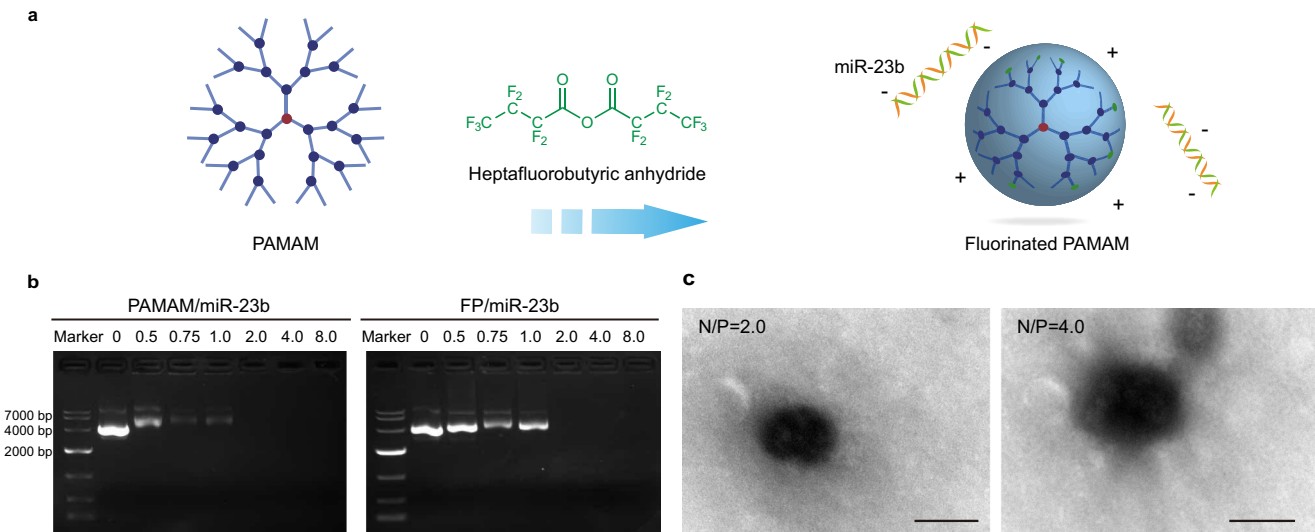

Fig. 1 | Synthesis and characterization of fluorinated PAMAM (FP). a The illustration represents the fluorination of PAMAM dendrimer through the modification with heptafluorobutyric anhydride and the formation of FP/miR-23b nanoparticles. b The binding and condensation ability of PAMAM and FP with miR-23b at different N/P ratios. c Representative transmission electron microscope (TEM) images of FP/miR-23b nanoparticles at N/P ratios of 2.0 (left) and 4.0 (right), respectively. Scale bar: 200 nm. A representative image of three biologically independent experiments from each group is shown in (b, c).

In addition, compared with PAMAM, the FP/miR-23b and HP/miR-23b nanoparticles appeared to be larger at the same N/P ratios, suggesting that PAMAM could assemble with the miR-23b plasmids into more condensed structure (Supplementary Fig. 6b). As shown in Fig. 1c, FP/miR-23b nanoparticles exhibited the spherical structure at N/P ratios of 2.0 and 4.0, with the diameter value of ca. 230.0 nm at N/P ratio of 2.0. Taken together, compared with unmodified PAMAM, the fluorination approach reduced the charge of FP/miR-23b nanoparticles and hampered the DNA binding and condensation, but the slightly loosen structure of the nanoparticles might be beneficial to release miR-23b in the cytosol.

## In vitro transfection efficiency of FP/miR-23b nanoparticles

First, the potential cytotoxicity of FP was tested in murine macrophage cell line RAW264.7 and bone marrow-derived macrophages (BMDMs) by MTT assay. As shown in Supplementary Fig. 7, FP demonstrated low cytotoxicity on these cells. To investigate whether fluorination promoted transfection, we examined the in vitro transfection efficiency of HP and FP, using the plasmid pEGFP-N3 as a model. As shown in Fig. 2a, b, the transfection efficiency of HP was strongly inhibited in different cell lines in comparison to unmodified PAMAM, indicating that the surface modification of butyric acid on PAMAM reduced the efficient transfection due to the lower positive charge of the nanoparticles. In this case, the transfection efficiency of FP was supposed to reduce since the surface charge of the FP nanoparticles also decreased after the fluorinated modification. However, in fact, FP demonstrated higher transfection efficiency than unmodified PAMAM at extremely low N/P ratios and exhibited comparable transfection ability with the commercial transfection reagent Lipofectamine 2000 in RAW264.7 cells and BMDMs. Concordantly, the phenomenon was observed in HeLa and NIH3T3 cells (Supplementary Fig. 8). These results indicated that although the fluorinated modification decreased the surface charge of PAMAM, the fluorine modified on PAMAM surface were essential in the efficient gene delivery to macrophages. Further, to examine the mechanism of FP's enhanced transfection, the endocytic pattern as well as the intracellular trafficking of FP/miR-23b nanoparticles were investigated in stimulated macrophages. As shown in Supplementary Fig. 9, FITC-labeled FP/miR-23b nanoparticles were successfully internalized into the cells with 38.4% uptake efficiency,

while low temperature could completely inhibit the cellular uptake, indicating the endocytosis relied on an energy-dependent manner. Meanwhile, a significant decrease of cellular uptake could be observed following the treatment with the chlorpromazine (24.2%), indicating that the clathrin-dependent endocytosis was mainly involved in the internalization of FP/miR-23b nanoparticles. In addition, the intracellular distribution was compared for PAMAM/miR-23b and FP/miR-23b nanoparticles following the successful internalization through confocal laser scanning microscopy (CLSM) (Fig. 2c). Clearly, the FP/miR-23b delivery efficiently escaped from the endosome after 4 h while most of PAMAM/miR-23b nanoparticles were entrapped in the acidic vesicles, indicating that the fluorination could improve the endosomal escape ability of nanoparticles. Moreover, the fluorination strategies have been widely applied in other cationic polymers such as polyethylenimine and poly(propylenimine) which endowed these polymers with both hydrophobic and lipophobic properties to facilitate the cellular uptake and endosomal escape of nanoparticles, thereby achieving enhanced gene transfection[46,47]. Considering that the similar transfection efficiency of FP could be achieved under N/P ratios of 2.0, 4.0, and 8.0 for the pEGFP-N3 transfection, lower N/P ratio should be chosen as an optimal ratio to avoid the cytotoxicity. Hence, the optimal N/P ratios of PAMAM and FP were adopted in the follow-up study, with value of 8.0 and 2.0, respectively.

The abundance of the macrophages residing in the inflamed synovium constantly secret the pro-inflammatory cytokines, resulting in the chronic inflammation and pannus in the inflamed joint. First, to mimic the inflammatory microenvironment in the arthritic joints, the BMDMs and RAW264.7 cells were exposed to different concentrations of TLR4 agonist lipopolysaccharides (LPS) for 24 h, aiming to confirm whether LPS could alter the intracellular miR-23b expression. Apparently, compared with non-stimulated macrophages, the expression of miR-23b was validated to decrease in response to different concentrations of LPS in both BMDMs and RAW264.7 cells, suggesting that miR-23b might be the potential regulator of the inflammatory response in macrophage (Supplementary Fig. 10). Simultaneously, the LPS-stimulated macrophages were transfected with different nanoparticles to check whether they could restore the miR-23b expression in activated macrophages. As expected, compared with non-transfected cells, free miR-23b and FP/negative control miRNA

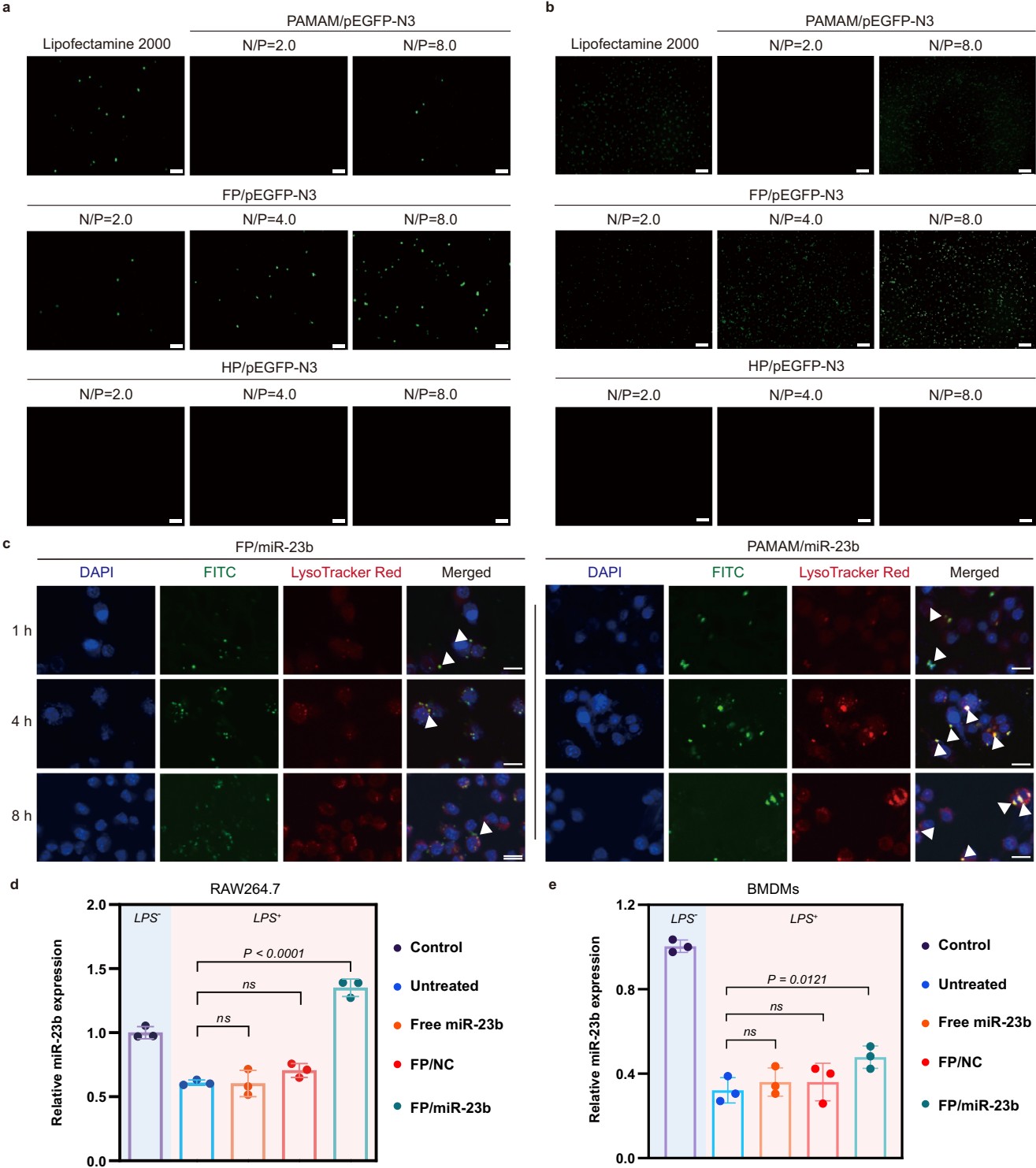

**Fig. 2 | In vitro transfection efficiency analysis of FP. a, b** Fluorescence images of the carriers-mediated transfection of pEGFP-N3 in LPS-stimulated BMDMs (**a**) and RAW264.7 cells (**b**) for 48 h. Lipofectamine 2000 was served as the control. Scale bar: 100 μm. **c** The representative images of the carrier-mediated miR-23b transfection in LPS-stimulated RAW264.7 cells. Nuclei, blue (DAPI); FP or PAMAM, green (FITC); lysosomes, red (LysoTracker Red). The white arrows represent the overlap of green and red fluorescence, indicating that the nanoparticles were entrapped in the lysosomes. Scale bar: 20 μm. A representative image of three biologically independent experiments from each group is shown in (**a–c**). **d, e** Relative miR-23b expression in LPS-stimulated RAW264.7 cells (**d**) and BMDMs (**e**) treated with different groups. LPS⁻ represents the cells without lipopolysaccharide stimulation, and LPS⁺ groups represent the cells treated with lipopolysaccharide (0.1 μg/ml) for 24 h. Data are presented as mean value ± SD (*n* = 3 independent experiments). One-sided statistical analysis is measured by one-way ANOVA with LSD test.

(FP/NC) nanoparticles did not alter the miR-23b expression level, while its expression was significantly increased after the FP/miR-23b transfection, indicating that FP could efficiently transfect miR-23b into the LPS-stimulated BMDMs and RAW264.7 cells (Fig. 2d, e).

## Inhibition of cell proliferation and inflammation by miR-23b delivery

Several studies have demonstrated that the transfection of miR-23b in tumor cells could trigger the in vitro cell death through the cell

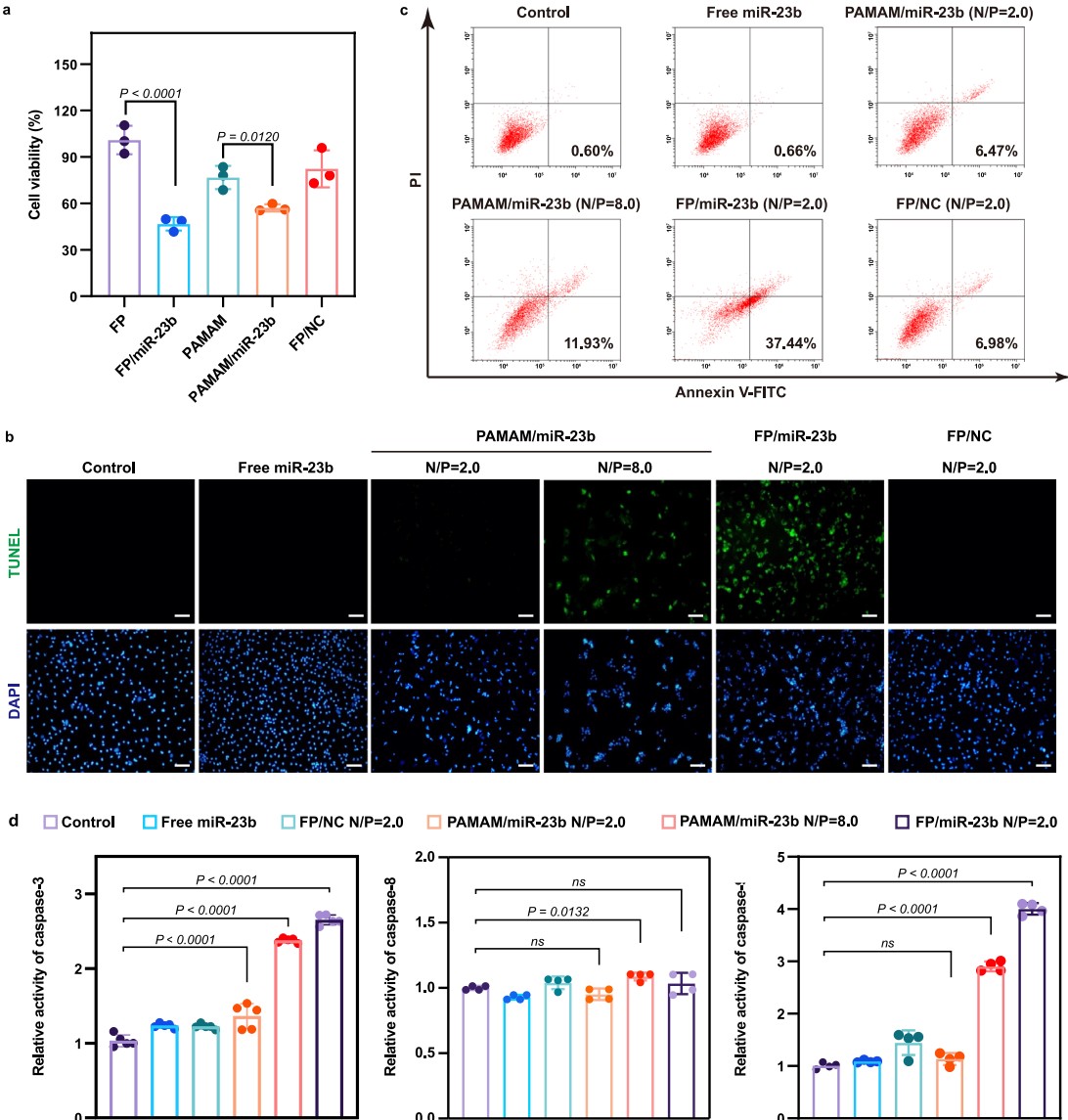

**Fig. 3 | Inhibition of cell proliferation after miR-23b transfection in LPS-stimulated BMDMs. a** The proliferative inhibition induced by the miR-23b transfection using MTT method. Data are presented as mean value ± SD (*n* = 3 independent experiments). One-sided statistical analysis is measured by one-way ANOVA with LSD test. **b** The apoptosis effect induced by miR-23b delivery visualized by TUNEL staining. Nuclei, blue (DAPI); TUNEL, green. Scale bar: 50 μm. A representative image of three biologically independent experiments from each group is shown. **c** The cell apoptosis analysis using the flow cytometry based on the Annexin V-FITC/PI staining, with apoptotic ratios of 0.60% in control group, 0.66% in the free miR-23b group, 6.74% in PAMAM/miR-23b (N/P = 2.0) group, 11.93% in PAMAM/miR-23b (N/P = 8.0) group, 37.44% in FP/miR-23b (N/P = 2.0) group and 6.98% in FP/NC (N/P = 2.0) group, respectively (*n* = 3 independent experiments). **d** Relative activity of caspase-3 (left), caspase-8 (middle) and caspase-9 (right) following the miR-23b transfection. Data are presented as mean value ± SD (*n* = 5 independent samples for caspase-3 and *n* = 4 independent samples for caspase-8 and caspase-9). One-sided statistical analysis is measured by one-way ANOVA with LSD test.

apoptotic pathway[22–24,36]. However, the similar antiproliferative effect of miR-23b in macrophages has been barely reported. To this end, we investigated whether miR-23b delivery could evoke the antiproliferative response using LPS-stimulated BMDMs and RAW264.7 cells as models. Notably, a significant inhibitory effect on the cell proliferation was observed after BMDMs were treated with nanoparticles harboring miR-23b, in which FP/miR-23b nanoparticles (N/P ratio of 2.0) exhibited a stronger antiproliferative response than PAMAM/miR-23b nanoparticles (N/P ratio of 8.0). Meanwhile, no obvious inhibition of proliferation was achieved for the FP/NC treatment (Fig. 3a). In addition, a higher percentage of dead cells could be

clearly visualized after the FP/miR-23b transfection in BMDMs cells using the live/dead cell staining, in which dead cells and viable cells emitted red and green fluorescence, respectively (Supplementary Fig. 11). The results were consistent with MTT analysis, from which we inferred that FP-mediated miR-23b transfection achieved the antiproliferative effect in LPS-stimulated macrophages. To further validate whether the proliferative inhibition was caused by the induction of apoptosis, the cells treated with FP/miR-23b nanoparticles were analyzed through TUNEL assay (Fig. 3b) and flow cytometric analysis based on Annexin V-FITC/PI staining (Fig. 3c). As depicted in TUNEL staining images, strong TUNEL signals were detected in BMDMs after

the transfection of FP/miR-23b and PAMAM/miR-23b (N/P ratio of 8.0) nanoparticles, indicating that the successful miR-23b transfection was able to trigger the cell apoptosis. Meanwhile, 37.44% of cells at early stage of apoptosis was obtained after the FP/miR-23b delivery whereas PAMAM/miR-23b (N/P ratio of 8.0) achieved a relatively lower apoptosis ratio (11.93%). Consistent with the results achieved in BMDMs, a similar apoptotic effect was observed in LPS-stimulated RAW264.7 cells (Supplementary Fig. 12a–d). Taken together, the miR-23b delivery could trigger the antiproliferative effect in LPS-stimulated macrophages by activating the apoptotic pathway, in which FP/miR-23b transfection was able to realize stronger apoptotic effect owing to its favorable transfection capacity. In the cell death signaling pathway, the caspase-related family has been identified to execute the cell apoptosis function, where the activation of caspase-3 could initiate the apoptotic program[48]. As shown in Fig. 3d and Supplementary Fig. 12e, the activity of caspase-3 dramatically increased following the miR-23b transfection in both LPS-stimulated BMDMs and RAW264.7 cells. Meanwhile, a significant increase in caspase-9 expression has also been observed after the treatment with FP/miR-23b and PAMAM/miR-23b (N/P ratio of 8.0) nanoparticles, while the expression level of caspase-8 appeared no significant changes. The activation of caspase-9 indicates the apoptosis is relied on the mitochondria-dependent apoptotic pathway whereas caspase-8 has been recognized as the initiator of the death receptor-mediated signaling pathway[49]. As noted, the FP/miR-23b transfection activated the mitochondrial-dependent apoptotic pathway, thereby leading to a cascade of apoptotic program in both LPS-stimulated BMDMs and RAW264.7 cells.

MiR-23b has been reported to be a significant inflammatory inhibitor, which was always downregulated in affected tissues of patients suffering from autoimmune diseases[19]. Thus, we hypothesized that the FP-mediated miR-23b delivery would relieve the inflammation in RA. In order to study the anti-inflammatory effect of FP/miR-23b nanoparticles, both BMDMs and RAW264.7 cells were treated by LPS for 24 h to stimulate the generation of pro-inflammatory cytokines such as TNF, IL1B and IL6, respectively. As shown in Fig. 4a–f and Supplementary Fig. 13a–f, the inflammation of BMDMs and RAW264.7 cells were significantly activated by LPS, as evidenced by the increased expression of pro-inflammatory cytokines. Notably, compared with FP/NC groups, these cytokines released by BMDMs and RAW264.7 cells significantly decreased after the treatment with miR-23b-harboring nanoparticles, suggesting that the miR-23b transfection in the LPS-stimulated macrophages could execute the anti-inflammatory effect. Meanwhile, the transfection of FP/miR-23b nanoparticles exhibited a stronger inhibitory effect on the inflammation than PAMAM/miR-23b group due to its superior transfection efficiency. To further discuss the underlying mechanism of anti-inflammatory effect induced by miR-23b delivery, the expression of miR-23b's potential targets (TGF-beta-activated kinase 1/MAP3K7 binding protein 2 (Tab2), Tab3 and IκB kinase alpha (Ikka)) was assessed after the miR-23b transfection. Previous reports have shown that LPS, as the classic agonist of TLR4 increased the expression of Tab2, Tab3, and Ikka in macrophages[50]. As expected, the TAB2, TAB3 and IKKA expressions dramatically increased in BMDMs and RAW264.7 cells after the LPS stimulation (Supplementary Fig. 14). As shown in Fig. 4g–i and Supplementary Fig. 15a–c, the mRNA level of Tab2, Tab3, and Ikka obviously decreased after the FP/miR-23b transfection in LPS-stimulated BMDMs and RAW264.7 cells. Consistent with the transcriptional level, the associated proteins were downregulated following the miR-23b delivery, suggesting that miR-23b could target TAB2, TAB3, and IKKA and further reduce their expression levels (Fig. 4j and Supplementary Fig. 15d). As reported, the TAB2/3 complexes recruited IKK complex to subsequently activate NF-κB signaling pathway, which could eventually lead to the inflammatory response[19]. Moreover, the secreted inflammatory cytokines particularly TNF, IL1B, and IL6 accelerated the activation of NF-κB signaling pathway to promote the production of

more inflammatory cytokines, which formed a feedback loop to strengthen the inflammation. Thus, it could be inferred that the downregulation of Tab2, Tab3, and Ikka by miR-23b would block the loop and further reduce the inflammatory effect. In summary, miR-23b could be efficiently delivered into the LPS-stimulated macrophages using FP as the carrier, which could trigger the apoptotic effect through the mitochondrial-dependent signaling pathway and simultaneously reduce the inflammatory response via inhibiting the expression of TAB2/3 and IKKA (Fig. 4k).

## Therapeutic effect of FP/miR-23b nanoparticles in experimental rheumatoid arthritis models

To evaluate the therapeutic efficacy of miR-23b in the RA treatment, the nanoparticles harboring miR-23b were intravenously administrated in an AIA rat model. The rats were subcutaneously injected with complete Freund's adjuvant to induce the inflammation, and the severity of arthritis could be scored by paw swelling and clinical indexes. The thickness change of hind paws was measured using a caliper to assess the inflammation in AIA rat model (Fig. 5a, b). Compared with healthy rats, an average increasing percentage of paw thickness could be observed in the arthritic rats received with saline and FP/NC nanoparticles, with the values of 124.6% and 120.2%, respectively, indicating that saline and NC miRNAs could not reduce the progression of RA disease. Since methotrexate (MTX) possessed anti-inflammatory effect in clinical use, it was utilized to treat RA at a dosage of 3.0 mg/kg which could achieve relatively slower inflammation progression than untreated control and saline groups. Notably, the FP/miR-23b delivery could inhibit the symptom of swelling in hind paws after 24 days post-immunization. Meanwhile, no obvious difference in the hind paw's thickness was observed for the MTX and FP/miR-23b groups at the end of therapy, demonstrating the ability of miR-23b to prevent the development of experimental RA disease. However, unsatisfactory therapeutic effect was obtained in PAMAM/miR-23b group owing to its limited transfection efficiency. At day 30 post-adjuvant administration, the thickness of paws significantly increased after the saline and FP/NC treatment, while the arthritic rats administered with FP/miR-23b nanoparticles exhibited comparable thickness to the healthy rats. Meanwhile, clinical indexes on fore paws and hind paws were assessed to monitor the severity of swelling and erythema (Fig. 5c), in which the arthritis-bearing rats in untreated group, saline group, and FP/NC group shared higher average indexes, illustrating the evidence of severe inflammation. However, the treatment with FP/miR-23b nanoparticles and MTX could relieve the inflammation with lower clinical indexes. Representative images of inflamed hind paws of arthritic rats receiving different treatments were captured through macroscopic observation (Fig. 5d), which was in accordance with the clinical index assessment. Except for the joint edema and erythema, the bone erosion and damage could also be observed in arthritis-bearing rats using the radiological analysis. Obviously, severe soft tissue swelling, marked destruction of tarsal bones and loss of joint space could be obtained in arthritic rats receiving saline and FP/NC, while moderate bone destruction was observed after the PAMAM/miR-23b treatment and limited bone erosion was developed following the MTX injection. Remarkably, the rats administrated with FP/miR-23b nanoparticles manifested a compact bone structure which was comparable to the healthy rats.

To further assess the bone function, the three-dimensional micro-CT bone reconstruction was conducted to validate whether miR-23b delivery could recover the bone destruction in AIA rats (Fig. 6a). In comparison with healthy rats, arthritic rats exhibited severe bone erosion, abnormal ossifications, and significant decrease in bone mineral density, particularly in saline, FP/NC and PAMAM/miR-23b groups. In contrast, bones of the rats receiving FP/miR-23b nanoparticles appeared less rough surface and more stable structure in ankle joints. Moreover, the bone mineral density (BMD) was used to

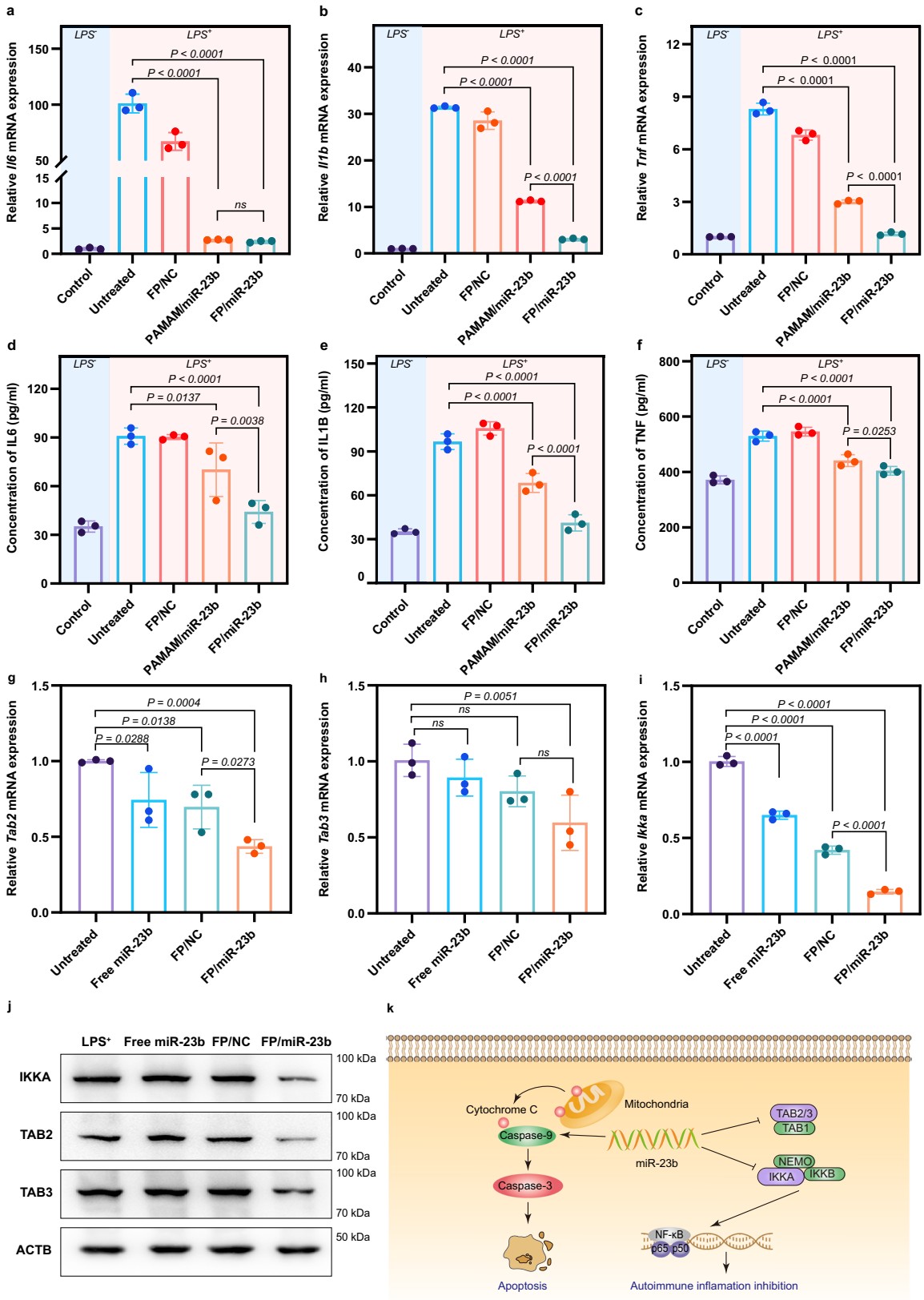

evaluate the bone formation, in which BMD of FP/miR-23b groups (5994.66 mg/cc) increased compared with the untreated group (5362.3 mg/cc), as shown in Supplementary Fig. 16. Besides, other parameters such as trabecular thickness, bone surface area/bone volume and trabecular spacing of FP/miR-23b groups, also returned to the normal level. These data demonstrated that FP/miR-23b delivery could efficiently protect the bones from destruction and erosion. Further, progressive inflammation could severely affect the bone function and subsequently influence the mobility of animals. Thus, the mobility of arthritic rats was evaluated through beam walking test (Fig. 6b, d) and footprint assay (Fig. 6c) following the treatment with different nanoparticles. In beam walking test, at day 7 post-adjuvant

**Fig. 4 | Inhibition of inflammation by miR-23b delivery. a–c** Relative mRNA expression of pro-inflammatory cytokines *Il6* (**a**), *Il1b* (**b**), and *Tnf* (**c**) in LPS-stimulated BMDMs after the miR-23b delivery. **d–f** Levels of IL6 (**d**), IL1B (**e**), and TNF (**f**) in cell culture supernatants were detected by ELISA methods. **g–i** Relative mRNA expression of *Tab2* (**g**), *Tab3* (**h**), and *Ikka* (**i**) in LPS-stimulated BMDMs cells after the miR-23b transfection. FP/miR-23b and FP/NC nanoparticles were prepared at N/P ratio of 2.0. **a–i** LPS⁻ represents the BMDMs without lipopolysaccharide stimulation, and LPS⁺ groups represent the BMDMs treated with lipopolysaccharide (0.1 μg/ml) for 24 h. PAMAM/miR-23b and FP/miR-23b (or FP/NC) nanoparticles were prepared at N/P ratios of 8.0 and 2.0, respectively. Data are presented as mean

value ± SD (*n* = 3 independent experiments). One-sided statistical analysis is measured by one-way ANOVA with LSD test. **j** The protein expression level of TAB2, TAB3, and IKKA in LPS-stimulated BMDMs after the miR-23b transfection. A representative blot of three independent experiments is shown. LPS⁺ represents the BMDMs treated with lipopolysaccharide (0.1 μg/ml) for 24 h. **k** The synergistic effect of apoptosis and anti-inflammation induced by miR-23b in macrophages. MiR-23b could trigger the mitochondrial-dependent signaling pathway to activate caspase-9 leading to the apoptotic effect, and the suppression of TAB2/3 and IKKA expression could block the NF-κB pathway and realize the inflammation inhibition.

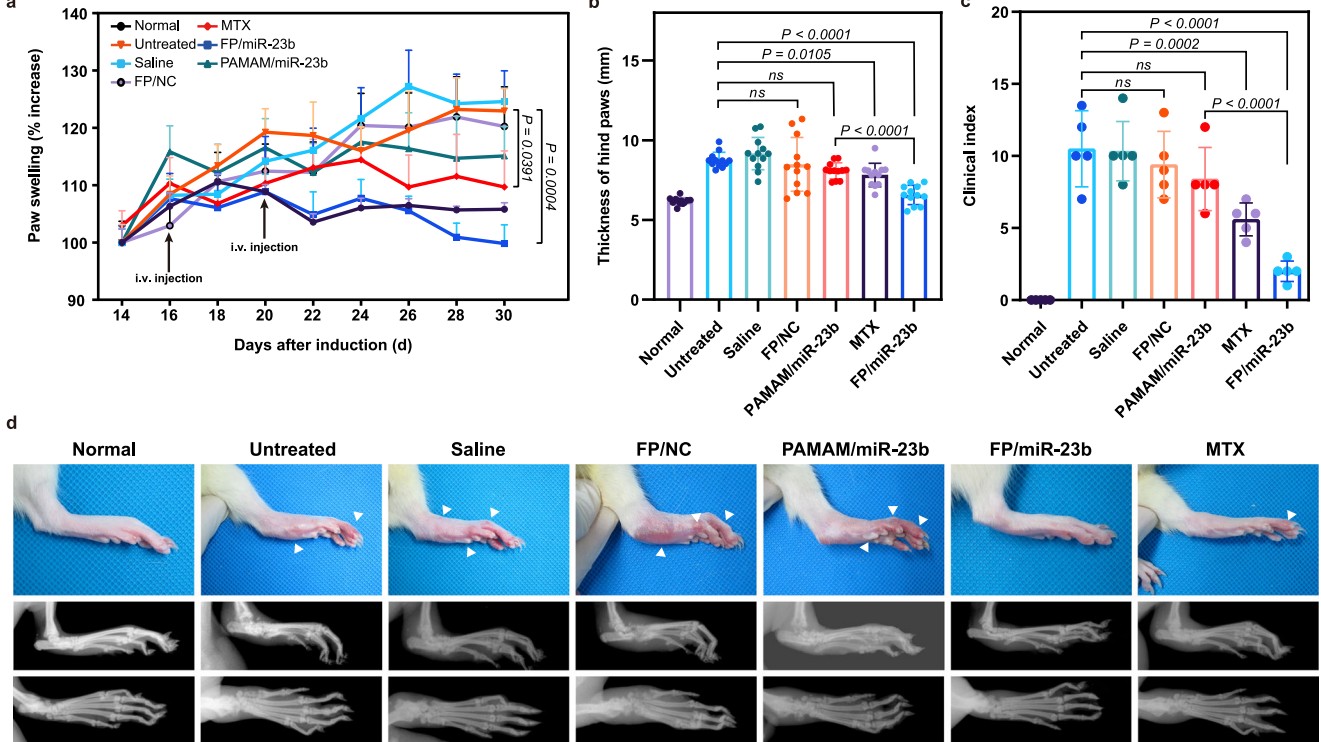

**Fig. 5 | Therapeutic efficiency of FP/miR-23b nanoparticles in AIA rat model in terms of changes in inflamed paws. a** The increase ratio of hind paw swelling during the period of 30 days post-adjuvant induction. Data are presented as mean value ± SD (*n* = 6 independent animals). **b** The thickness of hind paws assessed at day 30 post-adjuvant induction. Data are presented as mean value ± SD (*n* = 12 independent samples). **c** The clinical indexes of fore and hind paws assessed at the day 30 post-adjuvant induction. The clinical indexes were performed on 0–4 scale, where 0 = no signs of swelling or erythema; 1 = slight erythema and/or swelling;

2 = moderate edema; 3 = remarkable edema and limited use of the joint; 4 = excessive edema with joint rigidity. Data are presented as mean value ± SD (*n* = 5 independent animals). **a–c** One-sided statistical analysis is measured by one-way ANOVA with LSD test. **d** The macroscopic observations (upper) and radioactive examinations (lower) to assess the severity of soft tissue swelling and bone erosion at day 22 post-adjuvant induction. White arrows indicate the ankle swelling and redness. A representative image of three independent animals from each group is shown.

induction, all rats possessed the normal and stable motion ability to cross the entire beam within 3-4 s, while the arthritic rats began to spend double time to finish the beam walking at day 15 after immunization. The arthritis-bearing rats without treatment and those receiving saline and FP/NC nanoparticles could walk across the beam with an average time of above 10 s and started to craw with fore paws since the hind paws suffered from severe inflammation from day 19 after immunization. At day 21 post-adjuvant administration, arthritic rats receiving saline and FP/NC nanoparticles appeared to be tonic to barely keep the balance on the beam, in which the average time was measured to be over 15 s. Compared to these rats, the arthritic rats receiving PAMAM/miR-23b nanoparticles were able to cross the beam within 15 s. In addition, the rats maintained a good balance to cross the beam within 5 s in the MTX and FP/miR-23b groups at the end of therapy. The associated video was recorded at day 25 post-adjuvant administration (Supplementary Movie 1). Furthermore, motional function assessment was conducted through the footprint assay, in

which the stride length of hind paws was measured after the treatment. Compared with healthy rats, significant decrease in the stride length was observed in untreated group, saline group and FP/NC group. Particularly, at day 18 post-immunization, the rats in these groups suffered from the tonic joints and exhibited severe walking disability, which could not leave the footprint on the paper. In contrast, the arthritic rats treated with MTX and PAMAM/miR-23b nanoparticles exhibited relieved pain and partial motional recovery. In FP/miR-23b group, the rats set a blistering pace, indicating that the FP/miR-23b delivery could prevent the arthritic rats from walking disability and relieve the tonic pain.

Hyperplastic synovial tissue has been considered as one of the main characteristics in RA, which further led to the destruction of bone and cartilage. Thus, histopathological analysis was conducted based on the H&E staining of synovial tissue in ankle joints (Fig. 7a). In contrast to healthy rats, the arthritic rats exhibited an obvious infiltration of the inflammatory cells, pannus invasion and severe bone erosion of

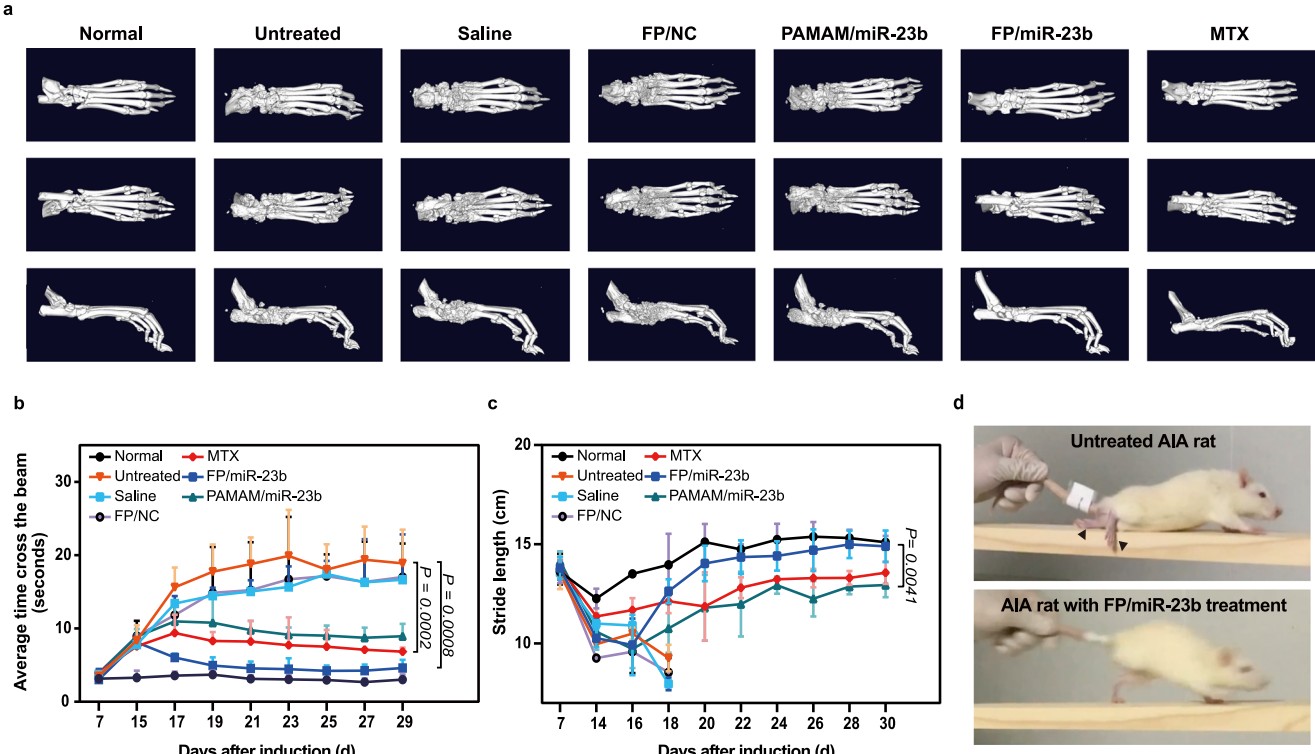

**Fig. 6 | Bone function and mobility analysis of AIA rats after the miR-23b delivery. a** The representative micro-CT images of arthritis rats after different treatments at day 22 post-adjuvant induction. **b** The beam walking test assessed during the period of 30-day post-adjuvant induction, in which the time to cross the 1-m beam was recorded. **c** Stride length measured by the distance of hind paws during the period of 30-day post-adjuvant induction. The rats in untreated, saline and FP/NC groups were void of walking ability, due to the tonic hind paws at day 18 post-adjuvant induction. **b**, **c** Data are presented as mean value ± SD ($n = 3$ independent animals), and one-sided statistical analysis is measured by one-way ANOVA with LSD test. **d** Beam walking test of AIA rats recorded at day 25 post-adjuvant induction, in which black arrows represent the tonic ankles. The video was recorded as Supplementary Movie 1.

joint tissues. After the treatment with miR-23b-containing nanoparticles, the infiltrates were reduced, and the bone erosion of ankle tissues was improved. Particularly, FP/miR-23b nanoparticles demonstrated a stronger inhibition of the hyperplasia of synovial tissues, as evidenced by the reduced number of cells in the synovium (Supplementary Fig. 17a). Meanwhile, we also graded the thickness of synovial tissues, inflammatory infiltration and pannus formation according to the histological examination (Supplementary Fig. 17b). Compared with untreated rats, the arthritic rats treated with FP/miR-23b nanoparticles exhibited lower histological synovitis scores (HSS) with statistical significance, which was highly in agreement with the observation of paw swelling. Furthermore, as shown in Fig. 7b, the articular cartilage was completely destroyed in the arthritic rats, while the cartilage was well preserved in rats received with MTX and FP/miR-23b nanoparticles. In accordance with our notion, the FP/miR-23b nanoparticles improved bone damage repair, as evidenced by the significant restoration of osteocalcin (OCN) in inflamed joints (Fig. 7c). These results indicated that the FP/miR-23b nanoparticles could alleviate the hyperplasia of the synovium to prevent the bone and cartilage destruction, therefore achieving the therapeutic efficacy.

Finally, the expression levels of *Tnf, Il1b*, and *Il6* in inflamed synovial tissue were assessed, since these cytokines secreted by the infiltrated cells including macrophages and FLSs were the key inflammatory cytokines in the development of RA, which could eventually stimulate the diverse pathological events to cause the joint damage. As shown in Fig. 8a–c, the mRNA expression of pro-inflammatory cytokines significantly increased in arthritic rats while no obvious decrease was observed after the treatment with saline and FP/NC nanoparticles. Remarkably, the PAMAM/miR-23b and FP/miR-23b nanoparticles could dramatically suppress the inflammatory expression in synovium,

in which FP/miR-23b group presented a similar anti-inflammatory response with the MTX treatment. Besides the pro-inflammatory cytokines in synovial tissue, these cytokines in serum were also detected. As shown in Fig. 8d–f, the increased pro-inflammatory cytokines in arthritis-bearing rats were strongly inhibited by the successful miR-23b transfection. Taken together, the efficient miR-23b delivery could realize the inhibitory effect on infiltrates in the synovium, and further diminish the pro-inflammatory cytokines secreted into the joint cavity to relieve the bone and cartilage destruction.

The renal and liver function after 12 h and 28 d post-administration of nanoparticles was measured to evaluate the acute inflammation response and the long-term toxicity, respectively. As shown in Supplementary Fig. 18a–e, the levels of serum biochemical parameters (ALT, AST, ALP, CREA, and BUN) in healthy rats were similar with those received with FP/miR-23b nanoparticles at 12 h and 28 d post-injection, indicating that FP did not induce the damage in the kidney and liver. Moreover, the section of main organs (heart, liver, spleen, lung, and kidney) was subjected to the H&E staining to evaluate the biocompatibility of nanoparticles (Supplementary Fig. 18f). In the staining images, there was no obvious organic lesion after the administration of FP/miR-23b nanoparticles. All these results demonstrated that the carrier FP possessed favorable biocompatibility in rats, which could be used as a promising candidate in future clinical use.

To further confirm the anti-arthritic effect of FP/miR-23b nanoparticles, the collagen-induced mice (CIA) model was established by the immunization of type II bovine collagen and incomplete Freund's adjuvant in the DBA1/J mice. Compared with AIA rats, the CIA mice shared more similar pathological and immunological features with RA patients such as the generation of autoantibodies towards collagen, symmetric joint involvement, synovitis, and cartilage and bone

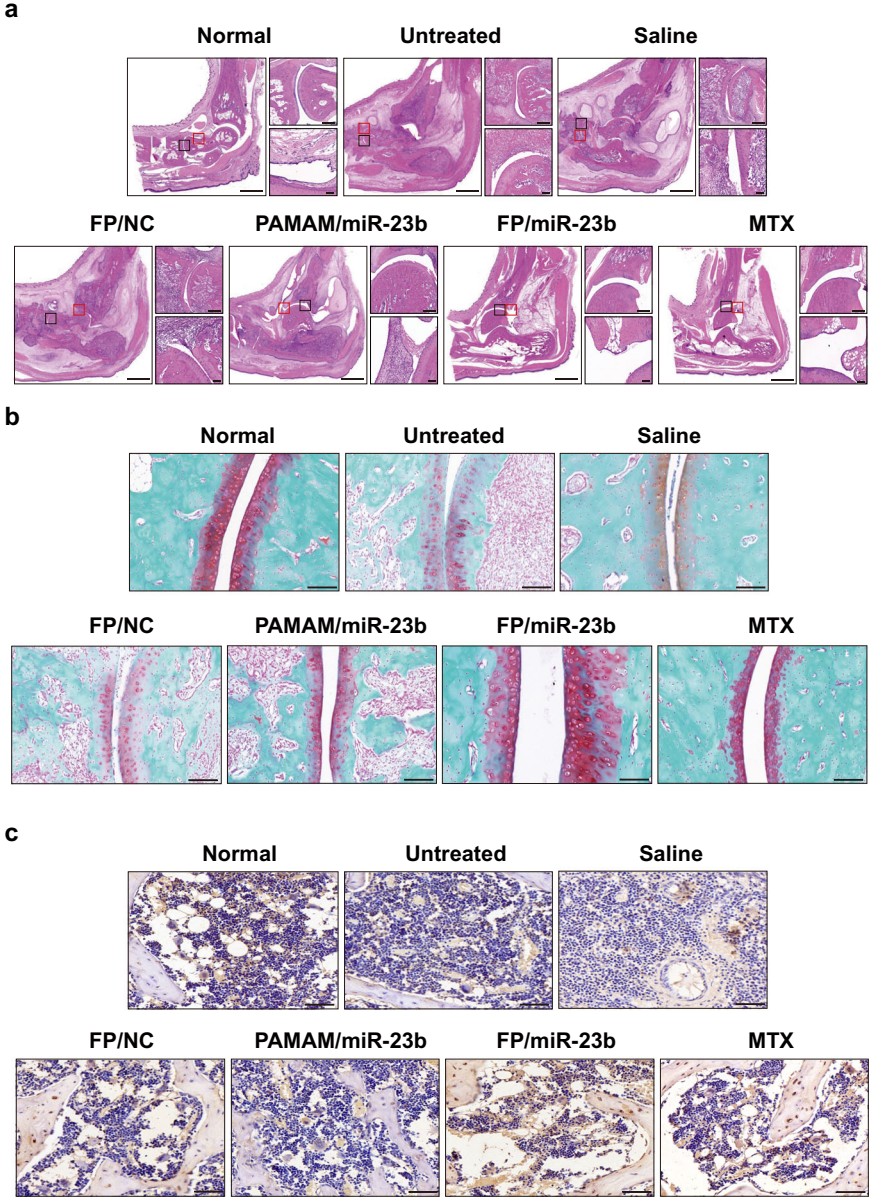

**Fig. 7 | FP/miR-23b nanoparticles reversed the bone erosion in AIA rats.**
**a** Histological change of ankle joints was analyzed by H&E staining. Scale bar: 2 mm. Black region represents the infiltrate of synovial tissues (upper right, scale bar: 200 μm). Red region represents the bone erosion (bottom right, scale bar: 50 μm). **b** Articular cartilages of ankle joints were identified by Safranin O/Fast Green staining. Scale bar: 50 μm. **c** Expression of the osteocalcin in arthritic joints was detected by immunochemistry. Scale bar: 50 μm. A representative image of three biologically independent samples from each group is shown in (**a**–**c**).

erosion[51]. As shown in Supplementary Fig. 19, the administration of FP/miR-23b nanoparticles significantly blocked the progression of disease after 28 days post-injection by characterizing the paw thicknesses, clinical scores, the typical RA symptoms and the mobility of CIA mice. In contrast, the FP-mediated delivery of NC miRNAs exhibited no therapeutic efficacy. Moreover, as demonstrated in Supplementary Figs. 20–22, FP/miR-23b nanoparticles reduced bone and cartilage erosion and cell infiltration of synovium, also exhibiting the therapeutic efficacy. We next examined the effect of FP/miR-23b nanoparticles on the secretion of pro-inflammatory cytokines in the synovial tissues and the blood. As shown in Supplementary Fig. 23, in both synovium and serum, the expression of TNF, IL1B and IL6 decreased in the arthritic joints of CIA mice after the administration of FP/miR-23b nanoparticles, suggesting that the nanoparticles could effectively control the arthritic inflammation. The potential side effects after the intravenous injection of FP/miR-23b nanoparticles were examined. As shown in Supplementary Fig. 24, the renal and liver

functions of CIA mice demonstrated no abnormalities after the treatment with FP/miR-23b nanoparticles for 12 h or 28 d. Besides, the histological analysis of major organs showed that FP/miR-23b nanoparticles did not induce obvious organic lesions, suggesting the desired biocompatibility of FP/miR-23b nanoparticles following systemic administration.

In addition, we also compared the therapeutic efficacy between HP/miR-23b and FP/miR-23b nanoparticles. As shown in Supplementary Figs. 25 and 26, the administration of HP/miR-23b nanoparticles did not demonstrate the alleviation of arthritic symptoms, with severe lesions of the joint and damaged articular tissues. Particularly, compared with FP/miR-23b nanoparticles, the HP-mediated miR-23b delivery barely reduced the expression of TNF, IL1B, and IL6, indicating that HP/miR-23b nanoparticles did not possess effective therapeutic outcomes. Thus, to figure out the therapeutic difference of HP/miR-23b and FP/miR-23b nanoparticles, we examined the miR-23b expression in the synovium after the administration of these nanoparticles.

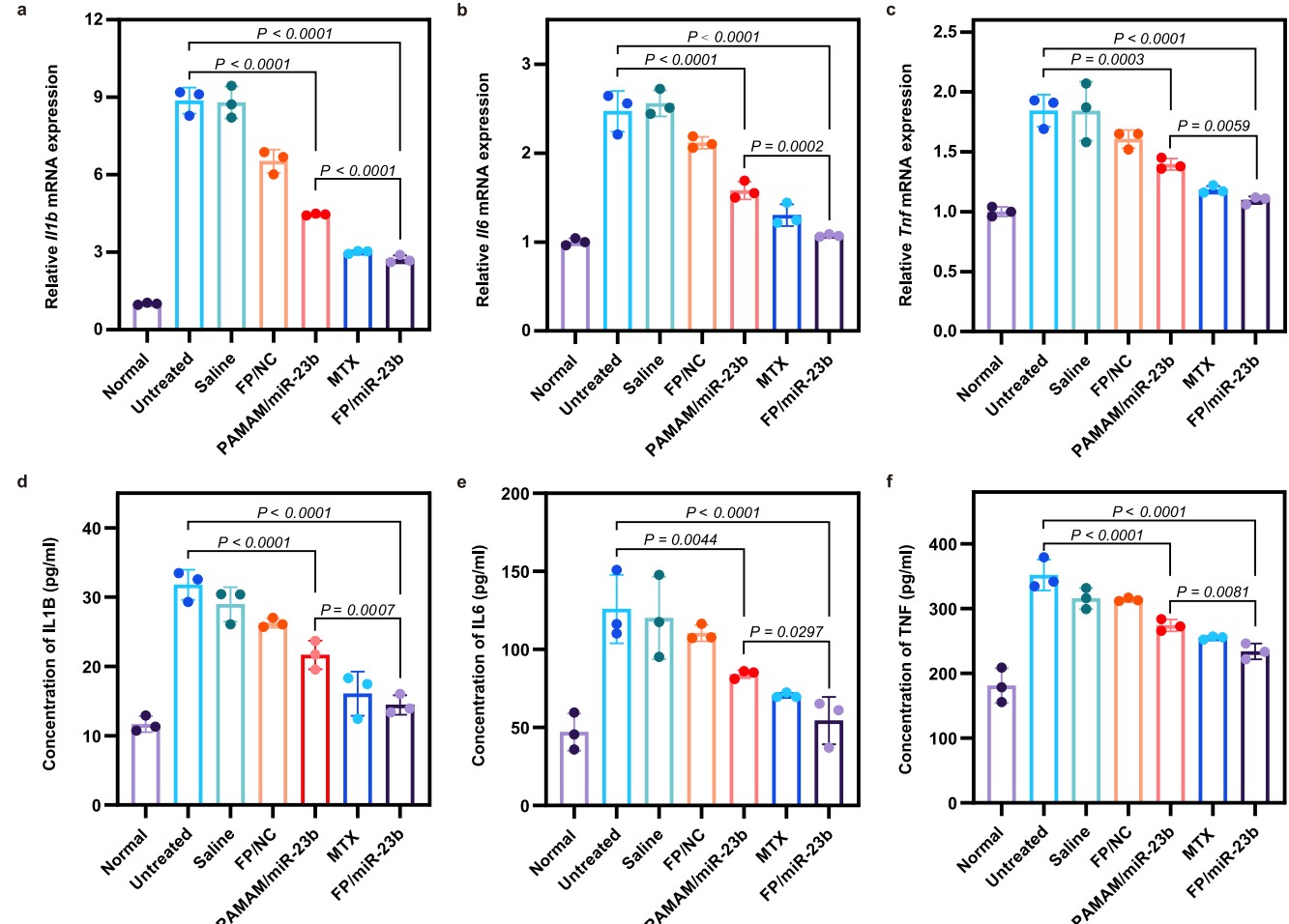

**Fig. 8 | Levels of pro-inflammatory cytokines in AIA rats. a–c** Relative mRNA expression of *Il1b* (**a**), *Il6* (**b**), and *Tnf* (**c**) in the inflamed synovial tissues measured by qPCR at day 23 post-adjuvant induction. The expression levels were calculated as the ratio of related cytokines to *Actb*. **d–f** Concentrations of pro-inflammatory cytokines in serum (IL1B (**d**), IL6 (**e**) and TNF (**f**)) detected by ELISA method at day 22 post-adjuvant induction. Data are presented as mean value ± SD (*n* = 3 independent animals). One-sided statistical analysis is measured by one-way ANOVA with LSD test.

As assumed, the HP-mediated miR-23b delivery did not improve the miR-23b expression in the synovial tissues in comparison with the untreated animals (Supplementary Fig. 27). This phenomenon was mainly attributed to the poor transfection efficiency of HP/miR-23b nanoparticles. Overall, it was evident that the miR-23b delivery could impede the development of experimental RA, in which FP/miR-23b nanoparticles exhibited superior therapeutic efficacy than PAMAM/miR-23b and HP/miR-23b nanoparticles.

**Accumulation of FP/miR-23b nanoparticles in the articular joints of CIA mice model**

To investigate the biodistribution of FP/miR-23b nanoparticles, we labeled miR-23b with Cy5 dye and tracked the fluorescence distribution of the FP/Cy5-labeled miR-23b nanoparticles in CIA mice after the intravenous injection (Fig. 9a–d). The greatest fluorescence of the nanoparticles was located in the liver and gradually decreased after 24 h, suggesting that the FP/miR-23b nanoparticles were captured and eliminated by the macrophages in liver tissue. Meanwhile, no lesion of liver tissues was observed after the administration of nanoparticles. Moreover, compared with free Cy5-labeled miR-23b, stronger fluorescence of FP/miR-23b nanoparticles was observed in joint tissues after 1 h post-injection. Specifically, after 24 h post-injection, the fluorescence of free miR-23b totally diminished in the inflamed joints while the fluorescence signal of FP/Cy5-labeled miR-23b nanoparticles still maintained visible. The quantitative analysis of fluorescence

intensity also exhibited that the signal of FP/miR-23b nanoparticles in the arthritic tissues was significantly higher than that of free miR-23b in multiple time points. These results demonstrated that the carrier FP could prolong the circulation time of miR-23b and preferentially mediate the accumulation in the arthritic joints through the passive targeting mechanism termed as ELVIS effect. Further, the FP/miR-23b nanoparticles exhibited the uniform distribution in the hyperplastic synovium of CIA mice after 1 h post-administration, which might be contributed to the leakage of nanoparticles through the invasive pannus in the synovial tissues (Fig. 9e). The hyperplastic synovium of RA was mainly composed of the synovial macrophages and fibroblasts; thus, we utilized the immunofluorescence to investigate the distribution pattern of the FP/Cy5-labeled miR-23b nanoparticles in the synovial macrophages and fibroblasts of synovium, as characterized by marking F4/80 and cadherin, respectively. As shown in Fig. 9f, the fluorescent colocalization of FP/Cy5-labeled miR-23b nanoparticles was simultaneously observed in both synovial macrophages and fibroblasts, indicating that the FP/Cy5-labeled miR-23b nanoparticles could be non-specifically captured by the cells residing in the synovium. Not surprisingly, although FP effectively increased the miR-23b delivery by enhanced endosomal escape effect, the fluorination strategy did not provide the gene carrier with the active targeting ability. Together, the FP/miR-23b nanoparticles could effectively deliver miR-23b into the inflamed tissues through ELVIS effect after the intravenous injection and accumulate in the hyperplastic synovium via the

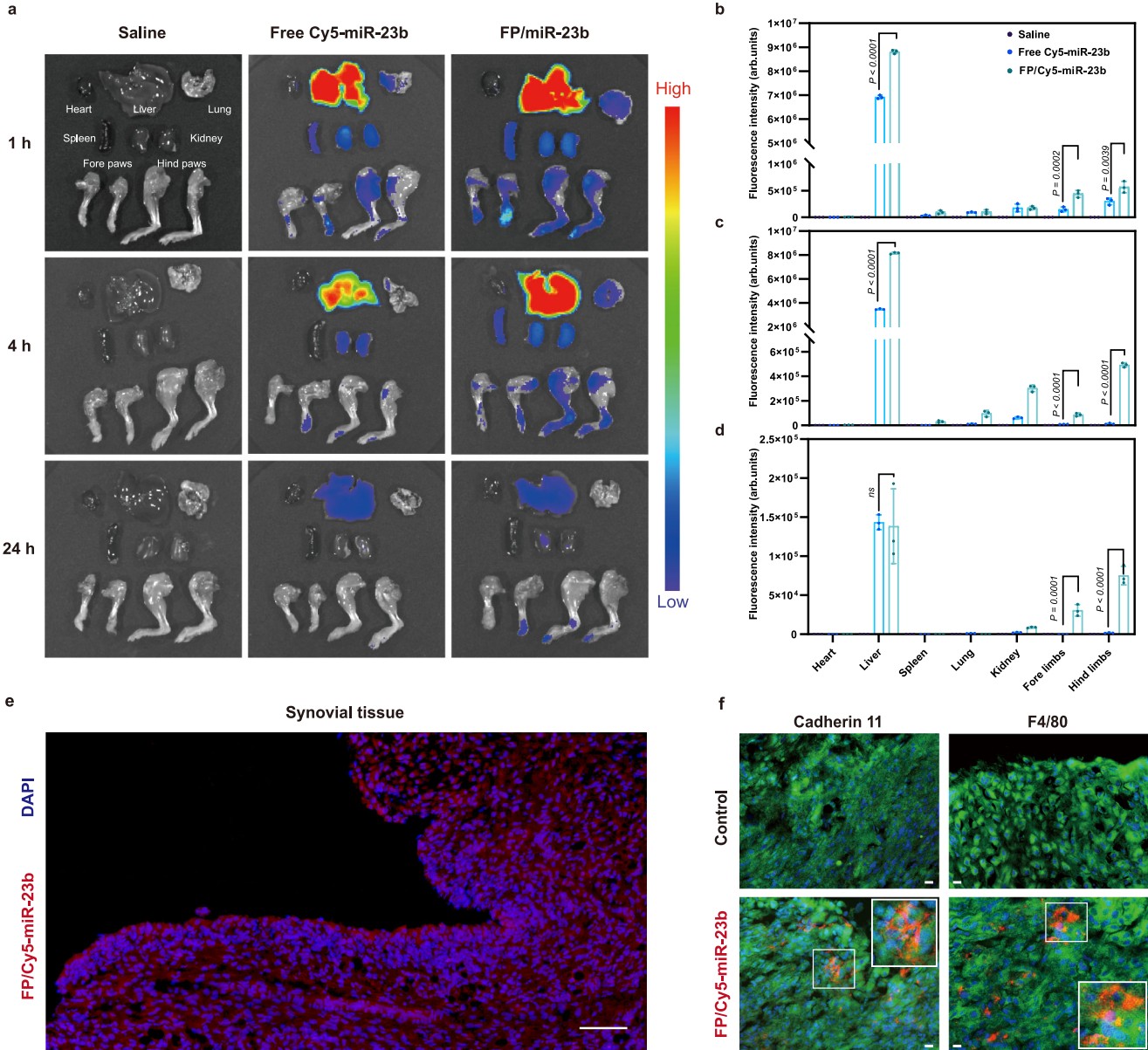

**Fig. 9 | The FP/miR-23b nanoparticles preferably accumulate in the arthritic joints of AIA rats. a** Ex vivo biodistribution of free Cy5-labeled miR-23b and FP/Cy5-labeled miR-23b nanoparticles in CIA mice at 1, 4, and 24 h post-administration. A representative image of three biologically independent samples from each group is shown. **b–d** The statistical quantification of fluorescence intensity of free Cy5-labeled miR-23b and FP/Cy5-labeled miR-23b nanoparticles in arthritic joints and major organs of CIA mice after the intravenous administration at **b** 1, **c** 4, and **d** 24 h, respectively. Data are presented as mean value ± SD (*n* = 3 independent experiments). One-sided statistical analysis is measured by one-way ANOVA with LSD test.

**e** Distribution of FP/Cy5-labeled miR-23b nanoparticles in the synovium of CIA mice. Nuclei, blue (DAPI); Cy5-labeled miR-23b, red. Scale bar: 100 μm. A representative image of three biologically independent samples is shown. **f** Distribution of FP/miR-23b nanoparticles in the synovial macrophages and fibroblasts. The macrophages and fibroblasts were detected by immunofluorescence analysis of F4/80 and cadherin 11, respectively. Nuclei, blue (DAPI); Cy5-labeled miR-23b, red; F4/80 or cadherin 11, green. Scale bar: 50 μm. A representative image of three biologically independent samples from each group is shown.

forming pannus, further leading to the endocytosis by synoviocytes to exert the therapeutic effect.

## In vivo anti-inflammation mechanism of FP/miR-23b nanoparticles

Inspired by the therapeutic efficacy of FP/miR-23b nanoparticles for the treatment of experimental RA, we further elucidated the in vivo mechanism underlying the therapeutic action of FP/miR-23b nanoparticles by determining the expressions of miR-23b and its related targets in inflamed joints at the end of the treatment. As shown in Supplementary Fig. 28, compared with the healthy animals, the miR-23b expression was downregulated in both arthritic models with

statistical significance. As expected, the miR-23b level significantly increased with the amelioration of inflammation in both AIA rats and CIA mice after the treatment with FP/miR-23b nanoparticles, indicating that the inflammatory response was mitigated by the miR-23b delivery. Thereafter we examined the expression of miR-23b targets in the synovium of AIA rats and CIA mice by qPCR and immunochemistry (Fig. 10a–d and Supplementary Fig. 29). Compared with the healthy animals, arthritic rats and mice exhibited elevated expressions of TAB2, TAB3, and IKKA in the synovial tissues, suggesting the activation of NF-κB signaling pathway. Consistent with the in vivo therapeutic results, the FP/miR-23b nanoparticles downregulated the expression of TAB2, TAB3, and IKKA in the synovium of inflamed joints.

Conversely, the FP/NC nanoparticles showed no effects on these targets. These results indicated that the miR-23b delivery inhibited the arthritic inflammation by suppressing the TAB2, TAB3, and IKKA expression in NF-κB signaling pathway. Besides, the immunochemical analysis of Ki67 and TUNEL was conducted to evaluate cell proliferation and apoptosis as a consequence of the restored miR-23b in the synovium. As shown in Fig. 10e and Supplementary Fig. 30, proliferating cells were abundant in the synovium of arthritic models while FP/miR-23b nanoparticles obviously inhibited the cell proliferation by triggering the cell apoptosis in the hyperplastic synovium, supporting the apoptotic effect of miR-23b. Moreover, as we noticed that the FP/miR-23b nanoparticles were accumulated in the liver tissue after the intravenous injection, we wondered whether the miR-23b delivery could affect the hepatic function. Thus, the liver tissues were collected at the end of the therapeutic process, followed by immunohistochemistry. As shown in Supplementary Figs. 31a–e and 32a–e, neither expressions of miR-23b targets nor apoptotic signal was significantly altered in arthritic animals after post-injection for couple of days, suggesting that the FP/miR-23b nanoparticles did not trigger the NF-κB signaling pathway as well as the induction of apoptosis in the liver. Besides, the hepatic miR-23b expression was evaluated by qPCR analysis, in which the FP/miR-23b group did not improve the miR-23b level in the liver tissue (Supplementary Figs. 31f and 32f). Thus, we assumed that the FP/miR-23b nanoparticles were gradually diminished by the liver tissues after the injection for several days, most of which were removed at the end of treatment since nanoparticles with size of ~200 nm were more easily scavenged by the hepatobiliary elimination[52]. Overall, our results indicated that the FP-mediated miR-23b delivery successfully increased the miR-23b expression in synovium and realized the synergistic effect by inducing the anti-inflammatory and antiproliferative responses for the treatment of experimental RA.

## Discussion

In the inflamed joints of RA patients, the synovial macrophages can spontaneously secrete pro-inflammatory cytokines leading to the pathogenesis of RA, which is vital in the progression of this autoimmune disease. Under the circumstances of the stimulation by the inflammatory network, the macrophages developed resistance to the cell death signal, resulting in the proliferation and accumulation in the inflamed synovium. The proliferative macrophages, in turn, could produce more cytokines, leading to the development of the vicious circle. As we hypothesized, the relationship between proliferation and inflammation is no longer individual but intimately connected, and thus the inhibition of either proliferation or inflammation is not efficient enough to prevent the disease progression. Previously, Zhu et al. demonstrated that miR-23b could suppress the inflammatory response in RA[19]. In addition, accumulating evidence has highlighted the function of miR-23b in the inhibition of tumor growth[22–24,36]. Thus, it will be likely that miR-23b could execute dual actions in the suppression of inflammation and proliferation in the RA treatment, as shown in Fig. 10f.

In this study, an efficient approach was developed to deliver miR-23b into inflamed joints, which could control the inflammation progression of RA through the inhibition of cell proliferation and inflammation. In gene therapy, dendrimer PAMAM has been widely used as a gene carrier in cancer treatment. However, a potential concern in the utilization of PAMAM in the RA treatment may be the limited transfection efficiency since the activated macrophages were difficult to be transfected. To this end, the fluorination strategy was employed to modify PAMAM since the fluorinated dendrimer has been tested to possess superior transfection efficiency in various commonly used cells. First, FP provided a binding ability to condense miR-23b into stable spherical nanoparticles with diameter of 230.5 nm at N/P ratio of 2.0. Next, FP exhibited much higher transfection ability in macrophages than PAMAM and Lipofectamine 2000 using pEGFP-N3 plasmid as a model. Although the charge density of FP/miR-23b nanoparticles decreased compared with the unmodified PAMAM, the fluorination approach still promoted the internalization of nanoparticles via the clathrin-dependent endocytosis route and provided a more rapid profile of endosomal escape to improve the transfection efficiency. In addition, the reduced surface charge density of dendrimer after the fluorination led to an easier unpacking pattern to release nucleic acids in the cytosol, which also accounted for effective gene transfection. Here, the macrophages were stimulated with LPS to evoke the inflammatory response, and intracellular expression level of miR-23b simultaneously decreased, suggesting that miR-23b could be a potential regulator of inflammation in macrophages. As anticipated, the transfection of FP/miR-23b nanoparticles into LPS-stimulated macrophages could achieve the restoration of miR-23b level, thereby executing its function in the suppression of inflammation and proliferation.

After the successful miR-23b transfection, the secretion of inflammatory cytokines has been obviously inhibited in the LPS-stimulated macrophages, demonstrating its function in the suppression of inflammation progression. The mechanism of miR-23b's effect on inflammation was identified to be associated with the efficient inhibition of TAB2/3 and IKKA, which could activate NF-κB signaling pathway. Through the block of NF-κB signaling pathway, extra inflammatory cytokines such as TNF, IL1B and IL6 could not be efficiently secreted into the joint cavity to activate the inflammatory response, which could eventually interrupt the inflammatory loop in the RA development. Besides that, stimulated by constantly secreted pro-inflammatory cytokines, the macrophages residing in the synovium exhibit a tumor-like behavior, generating a reduced sensitivity to the cell death signal. In cancer gene therapy, miR-23b has been confirmed to execute diverse physiological function in the inhibition of proliferation and metastasis of cancer cells. However, the antiproliferation effect of miR-23b was barely studied in autoimmune diseases including RA. Following the FP/miR-23b transfection, an obvious apoptotic effect was achieved in LPS-stimulated macrophages through mitochondrial apoptotic pathway, which was triggered by the activation of caspase-9. Previous reports have shown that the inhibition of NF-κB pathway could reduce the level of BCL2 family, leading to the loss of mitochondrial membrane potential[53], which was probably an important reason for the induction of cell apoptosis by miR-23b via the mitochondrial-dependent apoptotic pathway. Thus, the miR-23b delivery not only realized the inflammation suppression but also achieved an effective apoptotic response in progressively proliferative macrophages.

Subsequently, the miR-23b-containing nanoparticles were intravenously injected into the experimental RA models including AIA rats and CIA mice, and the therapeutic effect was monitored soon afterward, in which FP/miR-23b nanoparticles could significantly induce the inflammatory reduction and suppress the infiltration of synoviocytes in the arthritic joints. Meanwhile, the destruction and erosion of bone and articular cartilage in affected joints could be highly inhibited following the treatment with FP/miR-23b nanoparticles, thereby achieving the motional function recovery of arthritic models. Our data suggested that miR-23b contributed to the anti-arthritic effect in the experimental RA treatment.

After the intravenous administration, FP/miR-23b nanoparticles could be passively accumulated in the inflamed joints due to the presence of unique retention phenomenon namely ELVIS, and nonspecifically captured by synovial macrophages and fibroblasts in the hyperplastic synovial tissues. The miR-23b delivered in the synoviocytes targeted *Tab2*, *Tab3*, and *Ikka* to suppress the activation of NF-κB signaling pathway, thereby achieving the synergistic effect by anti-inflammatory and antiproliferative responses. Compared with the intra-articular injection, systemic administration of nanoparticles

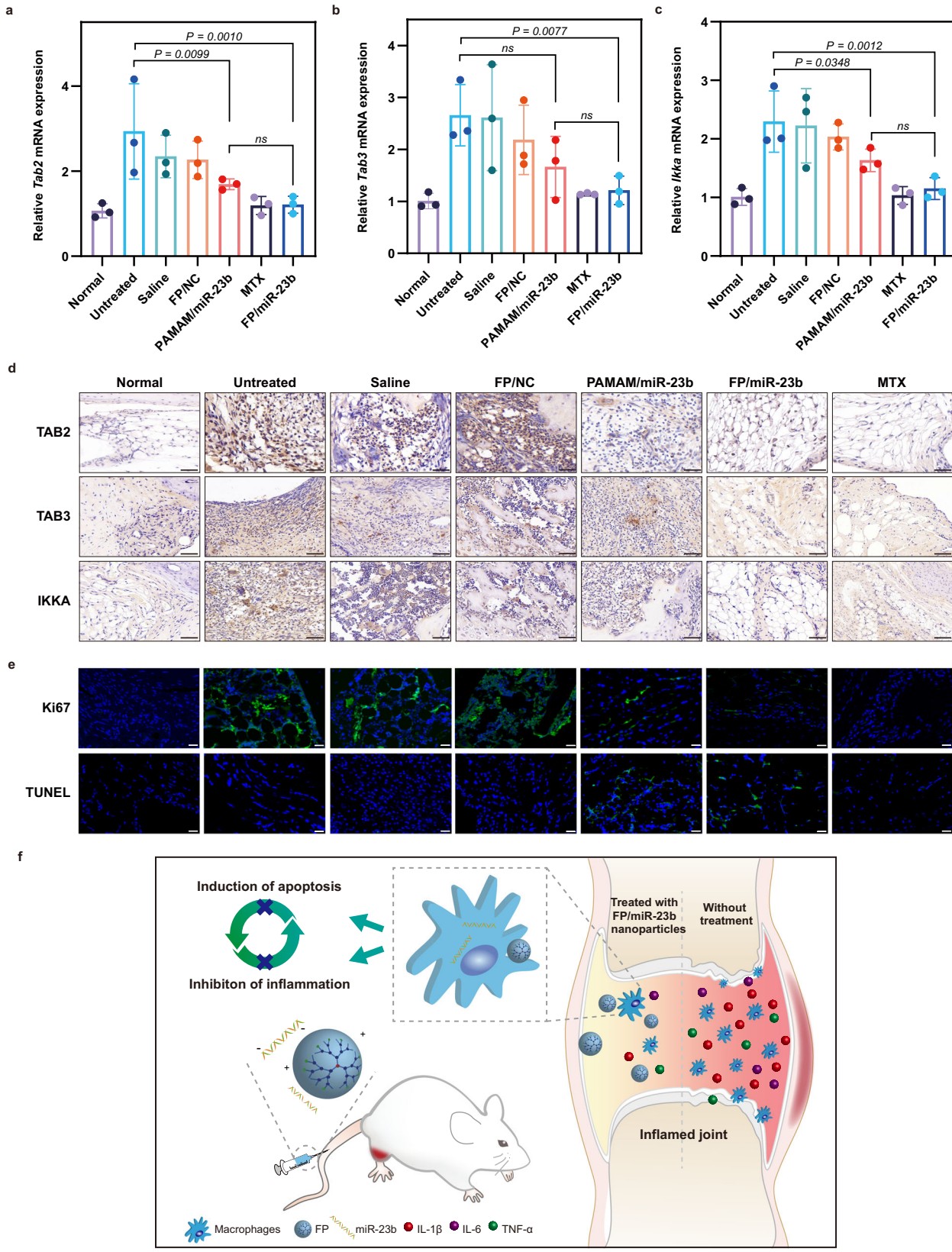

could easily satisfy the compliance of patients but might trigger unpredictable adverse effects. Of note, although the nanoparticles accumulated in the liver tissues after the intravenous injection, the nanoparticles did not elevate the hepatic miR-23b expression and cause obvious NF-κB activation as well as the apoptosis induction, which was possibly caused by the rapid elimination of nanoparticles by the liver[52]. In addition, we observed that FP/miR-23b did not cause acute renal and liver damage or generate long-term toxicity, suggesting that the FP/miR-23b nanoparticles were biocompatible to be used in systemic administration.

In conclusion, compared with the inhibition of certain pro-inflammatory cytokines, the miR-23b-based therapy substantially had

**Fig. 10 | The FP/miR-23b nanoparticles regulated the NF-κB signaling pathway to induce the anti-inflammatory and antiproliferative effect in the synovium of AIA rats. a**–**c** The mRNA expression of **a** *Tab2*, **b** *Tab3*, and **c** *Ikka* in the arthritic joints in AIA rats were measured by qPCR. Data are presented as mean value ± SD (*n* = 3 independent animals). One-sided statistical analysis is measured by one-way ANOVA with LSD test. **d** Levels of TAB2, TAB3, and IKKA in the arthritic joints of AIA rats were detected by immunochemistry. Scale bar: 50 μm. **e** The antiproliferative effect of FP/miR-23b nanoparticles in the synovial tissues of AIA rats by TUNEL and Ki67 immunofluorescence staining. Nuclei, blue (DAPI); TUNEL or Ki67, green. Scale bar: 50 μm. A representative image of three biologically independent samples from each group is shown in (**d**, **e**). **f** Schematic illustration of the intravenous administration of FP/miR-23b nanoparticles into AIA rats. Following the injection, the nanoparticles could be accumulated in the inflamed joints through ELVIS effect, and the released miR-23b executed its function after the internalization in synoviocytes. Finally, arthritis could be prevented through the induction of apoptosis as well as the inhibition of inflammation.

dual functions of inflammatory inhibition as well as apoptosis induction and focused on achieving a synergistic effect in the RA treatment. Moreover, based on the dual functions of miR-23b in the experimental RA treatment, the proof-of-concept trials to treat autoimmune diseases will be beneficial to restrict the recurrence of disorders.

## Methods
### Ethical statement
The animal experiments were performed according to the guidelines for the Care and Use of Laboratory Animal Experience of Jilin University and approved by the Institution Animal Ethics Committee of Jilin University (Changchun, China). (SY0208, SY0422, 2021SY0421, 2021SY0422, 2022YNPZSY0617, 2022YNPZSY0618).

### Materials
Amine-terminated polyamidoamine (PAMAM) dendrimer with the generation of 5.0 (ethylenediamine core) (CYD-150A) was purchased from Chenyuan Co. (Weihai, China). Heptafluorobutyric anhydride (H1006), butyric anhydride (W527211) were provided by Sigma-Aldrich (St Louis, MO). Methotrexate (MTX, MB1156) was purchased from Meilunbio (Dalian, China). The plasmid pEGFP-N3 was provided by Prof. Huayu Tian (Changchun Institute of Applied Chemistry, Chinese Academy of Sciences). The miR-23b (5′-ATCACATTGCCAGGGATTACC-3′) and negative control (NC) miRNA (5′-TTCTCCGAACGTGTCACGT-3′) were constructed into the plasmid vector by Gene Pharma (Suzhou, China), as shown in Supplementary Fig. 33. The plasmids were amplified in *Escherichia coli* DH5α and purified using Vigorous Plasmid Maxprep kit (N001) (Beijing, China). The plasmid expressing miR-23b were labeled by Label IT Nucleic Acid Labeling Reagents (MIR3700) purchased from Mirus Bio (Madison, WI). The primers used in the study was constructed by Sango Biotech (Shanghai, China) and their sequences are listed in Supplementary Table 1. Roswell Park Memorial Institute (RPMI) 1640 medium, Dulbecco's modified Eagle's medium (DMEM), fetal bovine serum (FBS), 1% penicillin/streptomycin and phosphate buffered solution (PBS, pH=7.4) were obtained from Gibco (Grand Island, NY). Mouse macrophage colony stimulation factor (mM-CSF) (96-315-02) was purchased from PeproTech (Rocky Hill, NJ). Red blood cell lysis buffer (C3702) was obtained from Beyotime (Nanjing, China). Lipofectamine 2000 and 3-(4,5-dimethylthiazol-2-yl)−2,5-diphenyltetrazolium bromide (MTT) were purchased from Invitrogen (Carlsbad, CA) and Amersco (Solon, OH), respectively. 4,6-diamidino-2-phenylindole (DAPI) (62248), RevertAid Premium Reverse Transcriptase (EP0441), LysoTracker Red (L12492), and LIVE/DEAD Viability/Cytotoxicity kit (L3224) for mammalian cells were acquired from ThermoFisher (Eugene, OR). Chlorpromazine (C0982) and genistein (G6649) were obtained from Aladdin (Shanghai, China) and Wortmannin was provided by J&K Scientific Co. (Beijing, China). TRNzol Universal reagent (DP424) for RNA isolation and Fast qPCR Master Mix (High Rox) (B639273) were purchased from Tiangen Biotech (Beijing, China) and Sangon Biotech (Shanghai, China), respectively. The complete Freund's adjuvant (CFA) (7009 and 7027), incomplete Freund's adjuvant (IFA) (7002) and type II bovine (20022) were purchased from Chondrex (Woodinville, WA). The following kits were used in these studies: one-step TUNEL apoptosis assay kit (C1086), Beyotime (Nanjing, China); Annexin V-FITC/PI apoptosis detection kit (BB4101) and BCA protein assay kit (BB3401), Bestbio (Shanghai, China); caspase-3 (G8090), −8 (G8200) and −9 (G8210) activity assay kits, Promega (Madison, WI); PrimeScript RT reagent kit (RR037A) and TB Green Premix Ex Taq (Tli RNaseH Plus) kit (RR820A), Takara (Dalian, China); rat TNF ELISA kit (RTA00), rat IL1B ELISA kit (RLB00), rat IL6 ELISA kit (R6000B), mouse TNF ELISA kit (MTA00B), mouse IL1B ELISA kit (MLB00C) and mouse IL6 ELISA kit (M6000B), R&D (Minneapolis, MN).

### Antibodies
In the western blotting assay, antibodies for TAB2 (3745, clone C88H10, lot 1, dilution 1:1000) and IKKA (2682S, lot 5, dilution 1:1000) were purchased from Cell Signaling Technology (Beverly, MA). Anti-TAB3 (ab124723, clone EPR5965, lot GR83293-9, dilution 1:5000), anti-ACTB (ab227387, lot GR3263411-3, dilution 1:5000) and horseradish peroxidase (HRP)-conjugated goat anti-rabbit IgG secondary antibody (ab6721, lot GR172025-4, GR297013-3, dilution 1:5000) were obtained from Abcam (Shanghai, China). In immunochemical analysis of joints, antibodies for TAB2 (NBP2-68833, lot R92304, dilution 1:1000) and cadherin 11 (NBP2-15661, lot 44034, dilution 1:250) were purchased from Novus Biologicals (Littleton, CO). Antibodies for TAB3 (PA5-116885, lot 3364EA11 and 35E8BA35, dilution 1:1000) were purchased from Invitrogen (Carlsbad, CA). Antibodies for IKKA (ab32041, clone Y463, lot GR117080-29 and GR3368230-15, dilution 1:1000), F4/80 (ab16911, clone BM8, lot GR3297595-4, dilution 1:50), Ki67 (ab16667, clone SP6, lot GR3375640-34, dilution 1:200) and horseradish peroxidase (HRP)-conjugated secondary antibody were acquired in Abcam (Shanghai, China). Anti-osteocalcin (23418-1-AP, lot GR3313195-53, dilution 1:100) was obtained from Proteintech (Chicago, IL). Peroxidase AffiniPure Goat Anti-Rabbit IgG (H + L) (111-035-045, lot 00000130634, dilution 1:500) and Alexa Fluor 488 goat anti-rabbit IgG secondary antibody (111-545-003, lot 000000101724, dilution 1:500) were acquired from Jackson ImmunoResearch (West Grove, PA). All commercially available antibodies have been validated by their manufacturer as indicated in their respective datasheet and/or website.

### Synthesis and characterization of FP
To synthesize the fluorinated PAMAM, the heptafluorobutyric anhydride was used as the fluorinated reagent[38]. Briefly, heptafluorobutyric anhydride (54.12 mg) and PAMAM (50.00 mg) were dissolved in 8 ml of methanol. The mixture was stirred at room temperature for 48 h, and then dialyzed against distilled water for another 48 h (MWCO: 3500 Da). The product was obtained through lyophilization and subjected to structural characterization using $^{19}$F NMR on AVANCE III spectrometer (Bruker, Rheinstetten, Germany). The molecular weight values of PAMAM and FP were measured by AB SCIEX 5800 MALDI-TOF mass spectrometer (Framingham, Massachusetts), in which 2,5-dihydroxybenzoic acid was used as a matrix. The butyric acid-modified PAMAM was synthesized as described above except for the usage of butyric anhydride (20.88 mg), and characterized using $^1$H NMR on AVANCE DMX 500 NMR spectrometer (Bruker, Rheinstetten, Germany). The morphology of FP/miR-23b nanoparticles were assayed by Hitachi H800 transmission electron microscope (TEM) at an accelerating voltage of 200 kV. The hydrodynamic diameter and zeta

potential of nanoparticles were measured by Malvern Nano ZS90 Zetasizer (Malvern, UK) and analyzed by Malvern Zetasizer software 7.11.

## Gel retardation assay

Nanoparticles were prepared by mixing 1 μg miR-23b plasmid and dendrimers in 20 μl distilled water at different N/P ratios. After the incubation at 37 °C for 30 min, they were analyzed using 1% (w/v) agarose gel electrophoresis (120 V, 20 min), and the bands were visualized on Universal Hood II system (Bio-Rad, USA). DNA maker of 10,000 bp (TaKaRa, Dalian, China) was used as the molecular weight scale. Uncropped scans of the gels are provided as the Source Data file.

## Bone marrow-derived macrophages generation

Bone marrow was isolated from the femur and tibia of thirty male C57BL/6J mice (15 ± 2 g, 4 weeks old) and filtered by passing a 70-μm cell strainer. The marrow was then incubated with red blood cell lysis buffer to remove blood cells and centrifuged at 250×g for 10 min to collect the precipitation of single cells. The cells were cultured in DMEM supplemented with 10% FBS, 1% penicillin/streptomycin and 50 ng/ml mM-CSF for 7 days, followed by the transfection procedure.

## Cultured cell lines

The murine RAW264.7 macrophages cultured in RPMI 1640 medium, and mouse NIH3T3 fibroblasts and human HeLa cells cultured in DMEM medium, were purchased from ATCC and cultured in the presence of 10% FBS in a humidified atmosphere with 5% $CO_2$ at 37 °C. Cells were routinely tested to be mycoplasma negative.

## In vitro transfection efficiency analysis

The cells were inoculated into 6-well plates at a density of $3.0 \times 10^5$ cells/well and cultured for 24 h, in which RAW264.7 cells and BMDMs were cultured in the presence of 0.1 μg/ml LPS. Then the transfection was conducted in FBS-free medium for 6 h using nanoparticles harboring 2.5 μg/ml pEGFP-N3 plasmid at various N/P ratios. After the culture in 10% FBS-containing medium for 48 h, the cells were subjected to observation under Olympus IX73P1F fluorescence microscope (Tokyo, Japan).

## Endocytosis mechanism analysis

Briefly, RAW264.7 cells were seeded into 6-well plates at a density of $3.0 \times 10^5$ cells/well and incubated with 0.1 μg/ml LPS for 24 h. Different endocytic inhibitors were used to treat the cells for 30 min, including 10 μg/ml chlorpromazine (clathrin-dependent endocytosis inhibitor), 150 μM genistein (caveolae-mediated endocytosis inhibitor), and 10 nM wortmannin (macropinocytosis inhibitor), and then the transfection of FITC-FP/miR-23b nanoparticles was conducted in FBS-free medium for 6 h. In addition, the transfection under 4 °C was used to check whether the endocytosis was performed in an energy-dependent manner. Finally, the harvested cells were washed with PBS three times and analyzed by FACSCalibur flow cytometry system (BD Bioscience, Franklin Lakes, NJ) equipped with CellQuest Pro software v6.0. The data were analyzed by Flowjo v10.0.

## Intracellular distribution analysis

The RAW264.7 cells were inoculated in six-well plates harboring coverslips ($2.0 \times 10^5$ cells/well) and stimulated with 0.1 μg/ml LPS for 24 h. The FITC-labeled nanoparticles (2.5 μg/ml miR-23b) were used to treat the cells for 1, 4, and 8 h, respectively, which were then stained with LysoTracker Red for 30 min. After washing with PBS three times, the cells were fixed by 75% ethanol and stained with DAPI solution for 30 min. Finally, the coverslips were used to obtain fluorescence images under LSM 710 confocal laser scanning microscope (Carl Zeiss Microscopy LLC, Jena, Germany).

## Inhibition of cell proliferation assay

Briefly, BMDMs and RAW264.7 cells were seeded in 96-well plates ($5.0 \times 10^3$ cells/well) and stimulated by 0.1 μg/ml LPS for 24 h. The PAMAM/miR-23b (N/P ratio of 8.0) and FP/miR-23b nanoparticles (N/P ratio of 2.0) with the miR-23b concentration of 2.5 μg/ml were added into the wells containing 200 μl FBS-free medium. After the incubation for 6 h, the cells were further cultured in 10% FBS-containing medium for an additional 48 h. Subsequently, 20 μl MTT solution (5 mg/ml) was added into each well, and the incubation was continued for 4 h. Finally, the formed formazan was dissolved by 150 μl dimethyl sulfoxide, and the inhibition of cell proliferation was calculated based on the ratio of optical density values of the treated and untreated cells measured at 492 nm on GF-M3000 microplate reader (Shandong, China).

## Cell apoptosis analysis

BMDMs and RAW264.7 cells were seeded in 6-well plates ($3.0 \times 10^5$ cells/well) and stimulated with 0.1 μg/ml LPS for 24 h. The transfection was conducted using different nanoparticles in FBS-free medium for 6 h, and the cells were further incubated in 10% FBS-containing medium for 48 h. For the apoptotic ratio analysis, the cells were treated with Annexin V-FITC and PI solutions according to the manufacturer's instructions and detected by CytoFLEX flow cytometry system (Beckman Coulter, Brea, CA). The data were analyzed by CytExpert 2.3. For the live/dead cell staining, the cells were washed with PBS three times and stained with 200 μl combined live/dead cell staining solution provided in the detection kit. For the TUNEL staining, the cells were fixed in 4% (w/v) paraformaldehyde solution for 30 min and then treated with 0.3% (v/v) Triton X-100 for 5 min, and then stained with TUNEL reagent at 37 °C for 1 h. After rinsing with PBS three times, the cells were subjected to the analysis on Olympus IX73P1F fluorescence microscopy (Tokyo, Japan) to acquire the images.

## Relative mRNA expression analysis in macrophages

The cell culture and miR-23b transfection were conducted as described in the section Cell apoptosis analysis. The total RNA of treated cells was extracted by TRNzol Universal reagent, and 2 μg RNA was utilized for the cDNA synthesis using PrimeScript RT reagent kit. Subsequently, 1 μg cDNA was applied for qPCR reaction by TB Green Premix Ex Taq (Tli RNaseH Plus) kit on ABI 7500 systems (Applied Biosystems, Foster City, CA). The relative expression levels of *Tnf, Il1b, Il6, Tab2, Tab3*, and *Ikka* were measured by $2^{-\Delta\Delta Ct}$ method using *Actb* as the reference gene. The sequences of primers used are listed in Supplementary Table 1.

## MiR-23b expression analysis in macrophages

The synthetic cDNA (1 μg) was obtained by the reverse transcription of total RNA using RevertAid Premium Reverse Transcriptase. Afterward, the qPCR was performed by SG Fast qPCR Master Mix (High Rox) (2×) on Applied Biosystems Stepone plus system (Applied Biosystems, Foster City, CA). The relative expression of miR-23b was normalized by $2^{-\Delta\Delta Ct}$ method using U6 gene as the reference. The primer sequences for miR-23b and U6 gene are listed in Supplementary Table 1.

## Measurement of cytokines in the cell culture supernatant

After the cells were transfected with miR-23b, the supernatants were collected by centrifugation at 400×g for 10 min to remove the cell debris. The concentration of TNF, IL-1β and IL6 in the supernatants were detected by the corresponding ELISA kits according to the manufacturer's instructions.

## Western blotting analysis

The cell culture and miR-23b transfection were conducted as described in the section Cell apoptosis analysis. The cellular lysates were obtained by the treatment of RIPA lysis buffer and collected by centrifugation at 14,000×g for 15 min. The concentration of extracted proteins in the supernatant were quantified using BCA protein assay kit

following the manufacturer's instructions. The proteins were separated by 12% SDS-PAGE and further electronically transferred into PVDF membranes. The membranes were blocked with 5% nonfat milk in a PBST solution at room temperature for 2 h and incubated with primary antibodies at 4 °C overnight, followed by the treatment with HRP-labeled secondary antibody at room temperature for 1 h. Finally, the bands were detected using ECL reagent and visualized by Tanon 2500 chemiluminescence imaging system (Shanghai, China). Uncropped scans of the blots are provided as the Source Data file.

## Animal
Male Lewis rats (200 ± 20 g, 5 weeks old) and male C57BL/6J mice (15 ± 2 g, 4 weeks old) were purchased from Beijing Vital River Laboratory Animal Technology Co., Ltd. (Beijing, China). Male DBA1/J mice (15 ± 2 g, 7 weeks old) were obtained from Shanghai SLAC Laboratory Animal Technology Co., Ltd. (Shanghai, China). All animals were housed in specific pathogen-free conditions at a temperature of 18–23 °C with 40–60% humidity under standard 12-h light/12-h dark cycle and allowed free access to food and water.

## Therapeutic treatment of adjuvant-induced arthritis (AIA) rats
Briefly, each Lewis rat (aged 6 weeks) was immunized subcutaneously with 100 μl complete Freund's adjuvant containing 5 mg/ml *Mycobacterium tuberculosis* at the base of the tail. After post-induction for ~12 days, rats appeared the symptoms of joint inflammation and pain, with swelling and redness of paws. Once the signs of inflammation were evident, 36 rats were randomly assigned to six groups ($n = 6$) as follows: Group 1 was healthy Lewis rats; Group 2 with arthritis received no therapeutic treatment was served as control; Groups 3–6 with arthritis were received an intravenous injection of saline, FP/miR-23b (N/P ratio of 4.0), FP/NC (N/P ratio of 4.0), PAMAM/miR-23b (N/P ratio of 4.0) and MTX (3.0 mg/kg body weight per injection), respectively. The AIA rats were intravenously injected with nanoparticles on day 16 and 20 after the adjuvant induction (miR-23b dose of 0.5 mg/kg body weight). The clinical parameters were monitored during the experiment, including the clinical index, mobility, the thickness of hind paws.

## Therapeutic treatment of collagen-induced arthritis (CIA) mice
Briefly, each DBA1/J mouse (aged 8 weeks) were subcutaneously immunized by 100 μg type II bovine collagen (2 mg/ml) emulsified in 50 μl CFA containing 2 mg/ml *Mycobacterium tuberculosis* at the base of the tail. On day 21, a booster injection with the collagen and IFA emulsion was administered into the mice to induce arthritis. Thirty-six mice were divided into six groups ($n = 6$ per group) as described above. The CIA mice were intravenously administered with the nanoparticles five times (once every other day) after 24 days post-primary immunization. During the treatment, the thickness of hind paws and the clinical indexes were monitored.

## Clinical assessment of arthritis
The clinical indexes were calculated for all four paws of each animal, and performed on a 0–4 score where score 0 = normal, no paw swelling or erythema; score 1 = mild swelling and/or erythema; score 2 = moderate swelling with edema; score 3 = remarkable edema and limited joint mobility; score 4 = severe edema and joint stiffness. The thickness of hind paws was measured with digital caliper.

## Beam walking test
A narrowed beam (5.0 cm in width, 1 m in length) was prepared and placed 25 cm above the platform. All animals were trained to walk across the beam before different treatments. The time of walking across the beam of each animal was then recorded by stopwatch after the induction of arthritis. Three consecutive runs were performed and the mean of three values was considered as the result for the rat on each testing day.

## Footprint assay
The plantar surface of each hind paw was painted with black ink and the rat was free to walk along a corridor (1 m in length, 10 cm in width) with a white paper on the base. Distance between hind paws' prints were measured with a digital caliper.

## X-ray and micro-computed tomography (micro-CT) analysis
The rats were euthanized after 22 days post-adjuvant induction. The hind paws of rats were dissected and subjected to X-ray analysis on Kodak In-Vivo FX PRO Imaging System (New Haven, CT, USA). After the X-ray assay, the hind paws were fixed in 4% (w/v) paraformaldehyde at 25 °C for 24 h. Similarly, the hind paws of CIA mice were dissected after 33 days post-primary immunization and stored in 4% (w/v) paraformaldehyde at 25 °C for 24 h. The collected hind paws were scanned in 35-μm isotropic resolution and analyzed by Quantum GX μ-CT (PerKinElmer, Ontario, Canada). Three-dimensional reconstructions were segmented using the materialize mimics 19.0 software (Materialize, Leuven, Belgium).

## Measurement of serum cytokines
The blood samples of AIA rats and CIA mice were collected on day 22 and 33 post-induction of arthritis, respectively, and allowed to clot at 25 °C for 30 min, followed by centrifugation at 1500×$g$ for 20 min. Afterward, the supernatant serum was isolated and used to detect the concentration of TNF, IL1B and IL6 by the corresponding ELISA kits according to the manufacturer's instructions.

## Expression-level analysis of pro-inflammatory cytokines and miR-23b targets in synovial tissue
The AIA rats were sacrificed on day 23 post-adjuvant induction. The CIA mice were sacrificed on day 33 post-primary immunization. The animals were dissected to obtain synovitis of joint tissues. The total RNA of tissues was extracted in TRNzol Universal reagent by vigorous mechanical shaking. The cDNA synthesis was performed using PrimeScript RT reagent kit and subjected to qPCR reaction using TB Green Premix Ex Taq (Tli RNaseH Plus) kit on ABI 7500 systems (Applied Biosystems, Foster City, CA). The relative mRNA expression was normalized by $2^{-\Delta\Delta Ct}$ method, and the primer sequences for *Tnf, Il1b, Il6, Tab2, Tab3, Ikka,* and *Actb* are listed in Supplementary Table 1.

## MiR-23b expression analysis in synovial tissue
The cDNA synthesis and qPCR reaction were conducted by RevertAid Premium Reverse Transcriptase and SG Fast qPCR Master Mix (High Rox) (2×), respectively. Each sample was run in triplicates on Applied Biosystems Stepone plus system (Applied Biosystems, Foster City, CA). The relative expression of miR-23b was normalized by $2^{-\Delta\Delta Ct}$ method using U6 gene as the reference, and the primer sequences of miR-23b and U6 gene are listed in Supplementary Table 1.

## Histopathological analysis of joints
The ankle tissues of arthritic models were collected, fixed in 4% (w/v) paraformaldehyde at 4 °C for 24 h and decalcified with 10% (w/v) EDTA solution at 25 °C for three weeks. Afterward, the joints were embedded in paraffin and sectioned into 30 μm-thickness slices for hematoxylin and eosin (H&E) staining and Safranin O/Fast Green staining, which were then observed using Olympus DP70 microscopy (Tokyo, Japan). The infiltrated cells were quantified by Image J software (v1.5j8, National Institutes of Health, Bethesda, MD). The histological synovitis scores (HSS) of hyperplastic synovium were assessed according to Supplementary Table 2.

## Immunochemical analysis of joints
The embedded joint tissues were sectioned into 10 μm-thickness slices and then applied for the antigen unmasking in EDTA solution (Daixuan, Shanghai, China) for 12 min under the microwave. The hydrogen

peroxide blocking was conducted using 3% (v/v) hydrogen peroxide solution for 25 min in the dark. Afterward, the slides were blocked with 3% (w/v) BSA solution diluted in PBS buffer for 30 min and then incubated with individual primary antibody at 4 °C overnight. The immunochemical analysis was conducted by applying HRP-conjugated secondary antibody at room temperature for 1 h, followed by the visualization of diaminobenzidine (DAB) substrate and hematoxylin. Finally, the sectioned tissues were rehydrated in ethanol and xylene, and mounted by neutral balsam. The slides were observed by Olympus DP70 microscopy (Tokyo, Japan).

### Antiproliferative analysis of joints
After the treatment of the arthritic animals, the ankle tissues were dissected for the section. The parafilm-embedded joint tissues were sectioned for TUNEL and anti-Ki67 staining, and the nucleus was stained by DAPI. The slides were observed by Olympus DP70 microscopy (Tokyo, Japan).

### Biodistribution analysis
The miR-23b expressing plasmids were labeled with Cy5 dye using Label IT Nucleic Acid Labeling Reagent and then purified by ethanol precipitation. The CIA mice were intravenously injected with saline, free Cy5-labeled miR-23b, and FP/Cy5-labeled miR-23b nanoparticles. The dosage of miR-23b was 0.5 mg/kg. The mice were sacrificed, and their organs and inflamed joints were dissected. The ex vivo fluorescence of tissues ($\lambda_{ex}$: 640 nm and $\lambda_{em}$: 680 nm) were observed by Caliper IVIS Lumina III In Vivo Imaging System (PerkinElmer, USA).

### Immunofluorescence staining of joints
The CIA mice were intravenously injected with FP/Cy5-labeled miR-23b nanoparticles for 1 h. Afterward, the CIA mice were sacrificed, and their ankle tissues were collected for sections. The tissues were sectioned into 10-μm slides, blocked with 10% rabbit serum at 37 °C for 30 min, and incubated with anti-F4/80 and anti-cadherin 11 antibodies at 4 °C overnight, respectively. After washing with PBS four times, the slides were incubated with Alexa Fluor 488-labeled secondary antibody and stained with DAPI for the nucleus. The fluorescent biodistribution of the FP/Cy5-labeled miR-23b nanoparticle was observed by Olympus DP70 microscopy (Tokyo, Japan).

### Biocompatibility analysis
Normal Lewis rats and DBA/1J mice were intravenously injected with nanoparticles with miR-23b dosage of 0.5 mg/kg body weight ($n = 3$), respectively. After 12 h and 28 days, the blood and main organs of the rats were collected. The serum was isolated as described above and the concentrations of aspartate aminotransferase (AST), alanine aminotransferase (ALT), alkaline phosphatase (ALP), creatine (CREA), and blood urea nitrogen (BUN) in the serum were measured by Olympus AU 400 automated clinical chemistry analyzer. The organs (heart, liver, spleen, lung, and kidney) were fixed with 4% (w/v) paraformaldehyde and then embedded in the paraffin. The organs were sectioned into 5-μm-thickness slices and examined by H&E staining. The images of slices were captured using IX73P1F fluorescence microscope (Olympus, Tokyo, Japan).

### Statistical analysis
Data were presented as mean value ± SD of triplicate experiments. Statistical analysis was measured by one-way ANOVA with LSD test to compare the values between multiple groups. All graphs and statistical analysis were performed by GraphPad Prism 8.

### Reporting summary
Further information on research design is available in the Nature Portfolio Reporting Summary linked to this article.

## Data availability
All data are available in the article, Supplementary Information files, or from the corresponding author upon request.  Source data are provided with this paper.

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

## Acknowledgements

We acknowledge the financial support from the National Key R&D Program of China to Q.L. (2018YFC1105401), the National Natural Science Foundation of China to Q.L. (81872928 and 32071267) and H.H. (32000897) and the Education Department of Jilin Province to H.H. (JJKH20211054KJ).

## Author contributions

H.H. designed and performed most of the experiments, analyzed data, and contributed to manuscript preparation. J.X. and W.C. assisted in the in vivo experiments, and J.J. conducted the mechanism analysis of miR-23b. Q.L. conceived, designed, and supervised the project and wrote the manuscript. All authors critically revised the manuscript and approved the submitted version.

## Competing interests

The authors declare no competing interests.
