## [Peer Review File · Nature Communications]

Fluorinated polyamidoamine dendrimer-mediated miR-23b delivery for the treatment of experimental rheumatoid arthritisEditorial Note: Figures presented by the authors in this file have been redacted at their request.

REVIEWER COMMENTS

Reviewer #1 (Remarks to the Author):

Han et al have utilized a new nanoparticle formulation to deliver miR-23b therapeutically to macrophages and arthritic rats via fluorinated dendrimers (FP/miR-23b). This approach reduced inflammatory cytokine production and increased cell death by Raw264.7 macrophages like cells lines in vitro. The data also show that established targets are repressed consistent with the effects occurring through direct targets of miR-23b. Reduced arthritis in response to administration of miR-23b via fluorinated dendrimers (FP/miR-23b) is also shown in a rat model by assessing clinical score, cytokines and histology, among other readouts. There is also evidence provided that FP/miR23b may be more potent than PAMAM/miR23b, while toxicity is not apparent based on the parameters tested.

Dendrimer-based miRNA delivery is an emerging non-viral approach, and this study provides evidence that it can be used to confer effective miR-23b delivery to macrophages and efficacy for reducing experimental arthritis symptoms in rats. Delivery of miRNAs in therapeutic settings remains an important endeavor, and this study provides some new data that dendrimer therapy may be efficacious.

However, it is worth noting that miR-23b replacement has been previously shown to be a promising approach to treating experimental arthritis, and dendrimers have been previously shown to enable miRNA replacement in vitro. So, the novelty here is that fluorinated dendrimers carrying miR-23b can reduce arthritis in rats (in vivo), and may be somewhat better than other dendrimer reagents in this context.

Thus, while the study does provide evidence for a potentially relevant miRNA delivery system in the context of arthritis, the overall novelty of this work is somewhat limited in terms of the biology of miR-23b. Further, the following points should be addressed.

Additional Comments:

1. In the introduction, it is mistakenly claimed that miRNAs regulate gene expression transcriptionally as opposed to post-transcriptionally (with the latter being the canonical mechanism of miRNA function). This should be addressed further (some limited evidence of direct transcriptional regulation by miRNAs) or corrected.
2. Grammar needs improvement throughout, and the results section should be more concise.
3. Too much reliance on Raw264.7 mouse macs for in vitro assays. Should use primary macs derived from mice or rats or human PBMCs (with this last option being the best).
4. Figure 7 Histology should be quantified.
5. miR23b delivery to specific cell types in joint tissues should be determined and quantified during in vivo experiments.
6. Ideally, the results from this study should be confirmed using an second, independent model of arthritis, perhaps in mice, because the in vivo efficacy of this approach is the strength of this manuscript.

Reviewer #2 (Remarks to the Author):

This manuscript describes very important/timely observations concerning the successful use of certain fluorinated PAMAM dendrimers as highly effective, low toxicity nanoscale delivery vectors for transporting critical si-RNA sequences to rheumatoid arthritis disease sites.

This manuscript is very well written, reasonably well referenced, very extensively documented with appropriate graphics and critical experimental support work to make all conclusions compelling and acceptable to this Reviewer.

As such, I recommend publication after first considering several comments, suggestions and revisions

as described below:

1. Page 21, line 458; Please identify the core of the G5; PAMAM dendrimer used in this work. Furthermore, Chenyuan Co. (Weihai, China) is not a wellknown source of high purity/quality PAMAM dendrimers. What characterization was performed on this dendrimer to assure the level of quality expected for this important investigation?
2. The term-PAMAM should appear in the title perhaps as follows: "Fluorinated PAMAM dendrimer-mediated miR-23 delivery for treatment of rheumatoid arthritis". Furthermore the term PAMAM should be used throughout the manuscript text to properly identify the critical dendrimer type used in this investigation.
3. The authors should include a brief discussion in the introductory section describing the anti-inflammatory properties observed earlier for simple functionalized PAMAM dendrimers [Ref.: A.S. Chauhan et al., Unexpected in vivo anti-inflammatory activity observed for simple surface functionalized Poly(amidoamine) dendrimers, *Biomacromolecules*, (2009), 10, 1195] and its relevance to the anti-inflammatory activity of simple fluorinated PAMAM dendrimers as described in this investigation.

Reviewer #3 (Remarks to the Author):

This manuscript describes the modification of 5th generation PANAM dendrimers which contain 128 terminal amines by perfluorobutyric anhydride. They claim the reaction modified approximately 48 of the 128 terminal amines and makes the modified dendrimer more hydrophobic.

They form a dendriplex with plasmid DNA that encodes the sequence for miR-23b. They characterize the complex and use it to transfect cells in culture and organs in vivo. They use a rat arthritis model and show data that indicates when the miR-23b plasmid nanoparticles were intravenously injected into the adjuvant-induced arthritis (AIA) rat model the RA symptoms with respect to joint swelling were eliminated and there was reduced bone damage and the rats recovered mobility.

Specific criticisms

1. Unfortunately, the authors do not appear to have used the correct control and hence are not able to conclude that the fluorinated dendrimer was more active than the unfluorinated PANAM 5th generation dendrimer because they did not use the butyric acid modified dendrimer in there experiments.

They used the unmodified 5th generation dendrimer as the control to deliver the miR-23b plasmid. I base my observation on supplemental figure 4 which indicates the zeta potential of the N/P 2/1 complex is +29 mV for the PANAM dendrimer but only +23 for the Fluorinated dendrimer. If the PANAM dendrimer had been butylated the 2/1 N/P complex should have had a similar zeta potential as the 2/1 fluorinated dendrimer plasmid DNA complex.

2. Line 119 , figure 1 b. The authors state that the "To verify the binding and condensation ability of FP with oligonucleotide, the derivative FP and miR-23b were incubated together at different N/P ratios and the formed nanocomplexes were evaluated using gel retardation assay"

Is this an oligonucleotide or a plasmid DNA? The figure needs a molecular weight scale on it. If the authors used both oligonucleotides and plasmid DNA in this manuscript they should state this in the methods and in the results where relevant. As it stands it is confusing to the reader.

3. Depending upon the core of the dendrimer, the 5th generation PANAM has a molecular weight =

28,800, if they attached 48 perfluorobutyric acids to the PANAM the MW should have been approximately 38,304. In supplemental Figure 2 they state their modified dendrimer has a molecular weight = 34,874, thus the number of amines modified would be closer to 31 not 48 as stated in sup Figure 2.

The authors should run a MALDI on the starting PANAM dendrimer so they can obtain a better estimate of the extent of modification. They should also estimate the number of fluorine groups on the dendrimer from the fluorine NMR of their modified dendrimer.

4. In addition, the 5th generation dendrimers used in these experiments are not optimized for gene delivery. The most active dendrimers have carboxylate defects in them. These defects significantly enhance their gene transfer active as reported in: Tang et al. *Bioconjug Chem* Nov-Dec 1996;7(6):703-14. doi: 10.1021/bc9600630.

5. The authors need to clarify these issues and if the appropriate control dendrimer that is the butylated 5th generation has not been used for the comparisons in the data this must be done.

Otherwise one cannot know if fluorinating the dendrimer has any beneficial effect on gene transfer compared to the butylated dendrimer.

Reviewer #4 (Remarks to the Author):

This manuscript present data concerning the development of FP/miR23b which was employed to intravenously treat AIA. While clinically FP/miR23b appeared to be effective, there are a number of concerns with the manuscript.

1) It is unclear how the in vitro transfection experiments relate to the in vivo effect. LPS-stimulated RAW cells are very different from the inflammation of AIA in vivo.

2) what are the effects of FP/miR23b on primary macrophages (in vitro developed from bone marrow or from human peripheral blood monocytes)?

3) many of the in vitro effects are artificial (cell lines and transfection and LPS), making it unclear what the mechanism for effectiveness in vivo is.

4) a description of the rats ability to walk are more than is needed to get the point across, while quantitation of histology and mechanistic studies of what is going on in the joint are not presented.

5) In figure 10, the authors provide a cartoon, suggesting the mechanism of action, but they present no in vivo data to support this mechanism, other than a reduction of cytokines. In fact, the data in Supp Fig 12, uses a t-test to show difference, but they mention ANOVA with a post-test, which is correct. What is "control" and why is FP/miR23b not different?

6) Specifically, does the FP/miR23b get into synovial macrophages or fibroblasts, is there increased apoptosis, is there an effect on NF-kb?

7) it appears the AIA was only performed in a single experiment and this needs to be repeated.

8) this model should not be referred to as RA, it is an experimental model of RA.

9) ELVIS is referred to in the abstract with no explanation of what this refers to until the introduction. If the authors want to suggest that their mechanism is working through this mechanism, they should define the parameters that permit them to make this claim.

Response to the comments from reviewers:

First, we would like to thank all the reviewers for their critical examination of our manuscript. Their comments are very helpful for revising the manuscript. We have made significant changes to address their concerns. The issues they raised, and our responses, are outlined below.

Reviewer #1 (Remarks to the Author):

Han et al have utilized a new nanoparticle formulation to deliver miR-23b therapeutically to macrophages and arthritic rats via fluorinated dendrimers (FP/miR-23b). This approach reduced inflammatory cytokine production and increased cell death by Raw264.7 macrophages like cells lines in vitro. The data also show that established targets are repressed consistent with the effects occurring through direct targets of miR-23b. Reduced arthritis in response to administration of miR-23b via fluorinated dendrimers (FP/miR-23b) is also shown in a rat model by assessing clinical score, cytokines and histology, among other readouts. There is also evidence provided that FP/miR23b may be more potent than PAMAM/miR-23b, while toxicity is not apparent based on the parameters tested.

Dendrimer-based miRNA delivery is an emerging non-viral approach, and this study provides evidence that it can be used to confer effective miR-23b delivery to macrophages and efficacy for reducing experimental arthritis symptoms in rats. Delivery of miRNAs in therapeutic settings remains an important endeavor, and this study provides some new data that dendrimer therapy may be efficacious.

However, it is worth noting that miR-23b replacement has been previously shown to be a promising approach to treating experimental arthritis, and dendrimers have been previously shown to enable miRNA replacement in vitro. So, the novelty here is that fluorinated dendrimers carrying miR-23b can reduce arthritis in rats (in vivo), and may be somewhat better than other dendrimer reagents in this context.

Thus, while the study does provide evidence for a potentially relevant miRNA delivery system in the context of arthritis, the overall novelty of this work is somewhat limited in terms of the biology of miR-23b. Further, the following points should be addressed.

Response: We sincerely thank you for taking the time to review the manuscript and pointing out the novelty of the study. We agree that the biological studies of miR-23b should be improved and its therapeutic mechanism should be clarified in more aspects. Thus, we conducted additional experiments on mouse bone marrow-derived macrophages (BMDMs) and collagen-induced arthritis (CIA) mice to validate the conclusions. Please find our itemized responses below.

Comment (C) 1: In the introduction, it is mistakenly claimed that miRNAs regulate gene expression transcriptionally as opposed to post-transcriptionally (with the latter being the

canonical mechanism of miRNA function). This should be addressed further (some limited evidence of direct transcriptional regulation by miRNAs) or corrected.

Response (R) 1: Thank you for your important comments. We are sorry for the inappropriate description. Indeed, miRNAs control the gene expression post-transcriptionally by hybridizing to targeted mRNA and thereby regulating their translation or stability (Filipowicz, et al. *Nat. Rev. Genet.* 2008, 9: 102; Lai, et al. *Nat. Genet.* 2002, 30: 363). As required, the description has been corrected. Please see line 61-62 in page 4 in the revised manuscript for details.

C 2: Grammar needs improvement throughout, and the results section should be more concise.

R 2: Thank you for your valuable suggestion. We have carefully checked the grammar errors throughout the manuscript. In addition, the revised manuscript has been proofread by a professional researcher in the field of nucleic acid delivery. Meanwhile, we have gone through the manuscript and corrected the ambiguous and tedious description of results in our revised manuscript. We really hope that the flow and the language level have been substantially improved.

C 3: Too much reliance on Raw264.7 mouse macs for *in vitro* assays. Should use primary macs derived from mice or rats or human PBMCs (with this last option being the best).

R 3: We agree with the reviewer's point that the primary macrophages derived from mice or rats, or particularly human peripheral blood mononuclear cells (PBMCs) are more convinced to elucidate the mechanism of miR-23b. Unfortunately, in the current situation of Chinese health system responding to COVID-19, it would not be justified to have healthy donors come in to take blood for the isolation of PBMCs alone at this difficult circumstance. Thus, we chose the mice bone marrow-derived macrophages (BMDMs) as an alternative to perform the *in vitro* study (Deng, et al. *Nat. Commun.* 2021, 12: 2174; Mayes-Hopfinger, et al. *Nat. Commun.* 2021, 12: 4546; Kobayashi, et al. *Nat. Commun.* 2016, 7: 11624).

For the revised version of the manuscript, we carried out new experiments to address the underlying mechanism on how miR-23b regulated the inflammatory and proliferative inhibition in BMDMs (**Fig. 3**, and **Fig. 4**). When the LPS-stimulated BMDMs were transfected with FP/miR-23b nanoparticles, we observed that the inflammatory response was significantly inhibited, as evidenced by the decreased content of cytokines (IL-1 β , IL-6 and TNF- α) (**Fig. 4a-f**). Notably, the expression of TAB2, TAB3 as well as IKK- α was inhibited by FP/miR-23b nanoparticles, suggesting that miR-23b was capable of suppressing the NF- κ B signaling pathway to realize its anti-inflammatory effect (**Fig. 4g-j**). As miR-23b has been proved to trigger the *in vitro* cell death through the cell apoptosis, we also tested the effect of FP/miR-23b nanoparticles on the proliferation of LPS-stimulated BMDMs by quantitative MTT assay, live/dead cell staining assay, TUNEL staining and flow cytometry analysis (**Fig. 3 and Supplementary Fig. 11**). We observed that the FP-mediated miR-23b delivery inhibited the proliferation of LPS-stimulated BMDMs by the induction of cell apoptosis. Meanwhile, the caspase-9 activity increased

in LPS-stimulated BMDMs after the transfection of FP/miR-23b nanoparticles, indicating that the cell apoptosis induced by miR-23b was dependent on the mitochondrial apoptotic pathway (**Fig. 3d**). These findings were consistent with the results obtained in RAW264.7 cells (**Supplementary Fig. 12, Supplementary Fig. 13, Supplementary Fig. 14, and Supplementary Fig. 15**).

Taken together, miR-23b was able to temper the inflammatory activation by suppressing the intracellular TAB2, TAB3 and IKK- α expressions in macrophages after the LPS stimulation. Besides, miR-23b has been demonstrated to possess a critical function in inhibiting the proliferation of macrophages by inducing the cell apoptosis through the mitochondria-dependent signaling pathway (**Fig. 3k**). These revised experiments achieved in BMDMs and corresponding results have been supplemented with appropriate discussion in the revised manuscript and Supplementary information. Meanwhile, the original results of the biological effects acquired in RAW264.7 cells have been presented in Supplementary information. Please refer to Fig. 3 and 4, Supplementary Fig. 10-15 and the highlighted version in Results (line 225-263, line 267-276 and line 281-290 in page 11-13), Discussion (line 547-553 in page 24) and Methods (line 675-681 in page 29-30 and line 751-755 in page 32).

C 4: Figure 7 Histology should be quantified.

R 4: Thank you for your kind suggestion. In our revised manuscript, we repeated the experiments on adjuvant-induced arthritis (AIA) rats and also supplemented the new data obtained in CIA mice. Accordingly, the lymphocytic infiltration and synovial hyperplasia of inflamed joints has been quantified by Image J software and simultaneously evaluated by histologic scores (**Supplementary Fig. 17, Supplementary Fig. 21 and Supplementary Table 2**). Compared with healthy rats, the hyperplastic synovium of arthritic rats was observed, with the increasing number of cells residing in the synovial membrane. After the intravenous injection of FP/miR-23b nanoparticles, the cell numbers in the synovium of AIA rats significantly decreased, indicating that the miR-23b delivery could limit the synovial hyperplasia. Moreover, the histologic scores of FP/miR-23b group were much lower than the untreated groups, indicating that FP/miR-23b nanoparticles could relieve the histologic feature and provide the therapeutic efficacy for AIA rats. Besides, we conducted the histologic quantification in the CIA mice after the treatment with FP/miR-23b nanoparticles. Similar results were obtained in CIA mice, in which FP/miR-23b nanoparticles significantly inhibited the histologic lesions in the inflamed paws (**Supplementary Fig. 21**). In conclusion, FP/miR-23b nanoparticles were proved to effectively inhibit the synovitis and protect the articular tissues, contributing to the therapeutic efficacy of experimental rheumatoid arthritis. Please refer to Supplementary Fig. 17, Supplementary Fig. 21 and the highlighted version in Results (line 380-390 in page 17) and Methods (line 844-851 in page 37).

C 5: miR-23b delivery to specific cell types in joint tissues should be determined and quantified during in vivo experiments.

R 5: We agree that the determination of delivered miR-23b to specific cells in joint tissues

will be necessary and helpful to better characterize the efficacy of nanoparticles as well as elucidate the therapeutic mechanism. Due to the difficulty of the measurement of plasmid's concentration and the complexity of metabolism *in vivo*, it was technically challenging for us to quantify the absolute amount of miR-23b in specific cell types of joints. Nevertheless, inspired by this comment and Reviewer #4's comment (point 6), we tracked the fluorescence of dye-labelled miR-23b in CIA mice and determined its distribution in joint tissues by immunohistochemistry to address the concern as much as possible. When CIA mice received the FP/Cy5-labelled miR-23b nanoparticles after 4 h post-administration, the nanoparticles partially accumulated in the inflamed joints, as evidenced by the higher fluorescence in joint tissues compared with those treated with free miR-23b (**Fig. 9a-d**). Meanwhile, we observed a significant increase of miR-23b expression in synovial tissues after the administration of FP/miR-23b nanoparticles (**Supplementary Fig. 25**). The articular distribution of FP/miR-23b nanoparticles may be attributed to their passive targeting ability through Extravasation through Leaky Vasculature and subsequent Inflammatory cell-mediated Sequestration (abbreviated as ELVIS) in the arthritic mice, which were consistent with the previous reports (Yuan, et al. *Adv. Drug Deliv. Rev.* 2012, 64: 1205; Xu, et al. *Nano Lett.* 2020, 20: 2258). Of note, the FP/miR-23b nanoparticles were mainly localized in the hyperplastic lining of synovium, which was probably caused by the leakage and penetration of nanoparticles through the invasive pannus in synovium (**Fig. 9e**). Since the synovial lining is mainly composed of the synovial macrophages and fibroblast-like synoviocytes (FLSs), we detected the fluorescence of FP/miR-23b nanoparticles in these two cells, where the synovial macrophages and FLSs were distinguished by the immunofluorescence of F4/80 and cadherin-11, respectively. As shown in **Fig. 9f**, the fluorescence of nanoparticles was observed in both macrophages and FLSs, indicating that the nanoparticles were non-specific to the cells residing in the hyperplastic synovium. Although the fluorinated dendrimer FP effectively increased the transfection efficiency of miR-23b by enhanced endosomal escape effect, the fluorination strategy did not improve the active targeting ability of the carrier. Thus, we reasoned that FP-mediated miR-23b delivery could achieve the accumulation of miR-23b in the inflamed joint tissues through ELVIS effect which could be further internalized by both synovial macrophages and FLSs lining on the synovial membrane. Please refer to Fig. 9, Supplementary Fig. 25 and the highlighted version in Results (line 452-484 in page 20-21), Discussion (line 591-596 in page 26) and Methods (line 867-881 in page 38).

C 6: Ideally, the results from this study should be confirmed using a second, independent model of arthritis, perhaps in mice, because the *in vivo* efficacy of this approach is the strength of this manuscript.

R 6: Thank you very much for your constructive comments. In our original manuscript, we have shown the *in vivo* effect of FP/miR-23b in AIA models. To address this reviewer's comment, we constructed the collagen-induced arthritis (CIA) mice as the second model of arthritis to validate the therapeutic efficacy of FP/miR-23b nanoparticles. Results of these studies were shown in Supplementary Fig. 19-27. Specifically, after the

intravenous administration of FP/miR-23b nanoparticles into the CIA mice, we observed that the nanoparticles inhibited the swelling of inflamed paws, protected the joints from bone erosion and improved the mobility of CIA mice (**Supplementary Fig. 19** and **Supplementary Fig. 20**). Moreover, the FP/miR-23b nanoparticles modulated the joint inflammation, reduced the hyperplasia of synovium and decreased the loss of cartilage (**Supplementary Fig. 21**, **Supplementary Fig. 22** and **Supplementary Fig. 23**). Next, the mechanism underlying the therapeutic action of FP/miR-23b nanoparticles was investigated. The nanoparticles could be selectively accumulated in the inflamed arthritic region by the passive targeting through ELVIS effect and located in the hyperplastic synovium due to the enhanced vascular permeability of invasive pannus (**Fig. 9**). The accumulation of FP/miR-23b nanoparticles significantly increased the miR-23b level in the synovial tissues and further inhibited the expression of TAB2, TAB3 and IKK- α , resulting in the suppression of inflammatory response in CIA mice (**Supplementary Fig. 25b** and **Supplementary Fig. 26**). Meanwhile, we observed that the delivery of miR-23b could significantly inhibit the cell proliferation in the synovium by the induction of cell apoptosis (**Supplementary Fig. 27**). Taken together, we concluded that the miR-23b delivery was able to attenuate the experimental rheumatoid arthritis by inhibiting the inflammatory response and the cell proliferation in synovial tissues. Please refer to Fig. 9, Supplementary Fig. 19-23, Supplementary Fig. 25-27, and the highlighted version in Results (line 424-450, 452-484 and 486-513 in page 19-24), Discussion (line 576-596 in page 25-26) and Methods (line 767-773 in page 33, line 787-795 in page 34 and line 844-881 in page 37-38).

Reviewer #2 (Remarks to the Author):

This manuscript describes very important/timely observations concerning the successful use of certain fluorinated PAMAM dendrimers as highly effective, low toxicity nanoscale delivery vectors for transporting critical si-RNA sequences to rheumatoid arthritis disease sites.

This manuscript is very well written, reasonably well referenced, very extensively documented with appropriate graphics and critical experimental support work to make all conclusions compelling and acceptable to this Reviewer.

As such, I recommend publication after first considering several comments, suggestions and revisions as described below:

Response: Thank you very much for your appreciation on the value of our work. We have carefully revised the manuscript according to the comments from you and the other three reviewers. Point-to-point responses to the comments are listed below.

C 1: Page 21, line 458; Please identify the core of the G5; PAMAM dendrimer used in this work. Furthermore, Chenyuan Co. (Weihai, China) is not a wellknown source of high purity/quality PAMAM dendrimers. What characterization was performed on this dendrimer to assure the level of quality expected for this important investigation?

R 1: Thank you for your kind suggestion. The core of PAMAM purchased from Chenyuan

is ethylenediamine. According to the reviewer's comments, we performed the structural analysis of the PAMAM dendrimer, including ^1H NMR and MALDI-TOF MS analysis. In addition, Chenyuan Co. kindly provided the results of HPLC and FT-IR of PAMAM. The HPLC chromatogram showed a single peak with retention time of 9.790 min, indicating that the PAMAM dendrimer was successfully achieved with high purity (**Fig. R1**). As shown in **Fig. R2**, ^1H NMR spectra of PAMAM showed six kinds of ^1H peaks, in which the four $-\text{CH}_2$ protons were assigned in the interior of the dendrimer (δH 2.35 ppm for a, δH 2.53 ppm for b, δH 2.65 ppm for c and δH 3.16 ppm for d) and two groups of $-\text{CH}_2$ protons were attributed to the outer layer (δH 2.53 ppm for b' and δH 3.22 ppm for d'). As shown in **Fig. R3**, the vibration band at 3295 cm^{-1} was contributed to the stretching vibration of $-\text{NH}-$ group, and the bands at 1642 and 1553 cm^{-1} were mainly ascribed to amide I and II. These results were consistent with the previous reports, indicating that the obtained PAMAM shared the identical structure with other PAMAM dendrimers (Hu, et al. *Int. J. Nanomed.* 2020, 15: 2751; Zhang, et al. *Anal. Methods* 2016, 8: 2218). As shown in **Supplementary Fig. 3**, although we did not obtain a line structure, a general molecular weight distribution was well observed, in which the molecular mass of PAMAM was determined to be 28859. This result was rather close to the theoretical molecular weight of 28826 based on the ideal structure, which indicated that PAMAM dendrimer used in the present study has intact skeleton containing 128 primary amines in the outer shell. Thus, the G5 PAMAM dendrimer obtained in Chenyuan Co. was identified with high quality and accurate structure, which could fulfill the requirements in the miR-23b delivery. Please refer to Supplementary Fig. 3 and the highlighted version in Methods (line 609-610 in page 26).

Editorial note: Figure redacted

Editorial note: Figure redacted

Editorial note: Figure redacted

C 2: The term-PAMAM should appear in the title perhaps as follows: “Fluorinated PAMAM dendrimer-mediated miR-23 delivery for treatment of rheumatoid arthritis”. Furthermore the term PAMAM should be used throughout the manuscript text to properly identify the critical dendrimer type used in this investigation.

R 2: Thank you for your kind suggestion. We are sorry for not clearly describing the dendrimer type in our initial submission. As you correctly pointed out, we now stated “PAMAM” to clearly identify the critical dendrimer type in the main manuscript. In addition, integrating this comment with the suggestion raised by Reviewer #4 (point 8), we have corrected the title of the manuscript as “Fluorinated polyamidoamine dendrimer-mediated miR-23b delivery for the treatment of experimental rheumatoid arthritis”.

C 3: The authors should include a brief discussion in the introductory section describing the anti-inflammatory properties observed earlier for simple functionalized PAMAM dendrimers [Ref.: A.S. Chauhan et al., Unexpected in vivo anti-inflammatory activity observed for simple surface functionalized Poly(amidoamine) dendrimers, *Biomacromolecules*, (2009), 10, 1195] and its relevance to the anti-inflammatory activity of simple fluorinated PAMAM dendrimers as described in this investigation.

R 3: We sincerely appreciate your valuable comments. We have followed the reviewer’s

suggestion and discussed the anti-inflammatory effect of PAMAM dendrimer in the revised manuscript. It is a meaningful study conducted by Chauhan et al. that amine-terminated PAMAM dendrimer (G4) could inhibit the both cyclooxygenase (COX)-1 and COX-2 to achieve unexpected anti-inflammatory activity. Besides, Liang et al. demonstrated that the cationic nanoparticles scavenged the cell-free DNA (cfDNA) to relieve the RA symptoms, suggesting that non-viral gene vectors such as PAMAM dendrimers could be exploited to achieve the efficacious removal of DAMP molecules for RA treatment (Liang, et al. *Nat. Commun.* 2018, 9: 1). However, in the present study, neither FP/NC nor PAMAM/miR-23b nanoparticles have been found to exhibit the inherent anti-inflammatory response when CIA mice or AIA rats received those nanoparticles. We assumed that there were two probable reasons. Firstly, compared with amine or hydroxyl terminated PAMAM dendrimers, the anti-inflammatory response diminished for the dendrimers presenting perfluoro acid groups. Secondly, in the study conducted by Chauhan et al., the rats were daily received with G4 PAMAM at a dose as high as 8 mg/kg in an intraperitoneal administration. In contrast, we intravenously injected approximately 2.6 mg/kg nanoparticles twice into AIA rats during the treatment, and thus it was postulated that FP/NC and PAMAM/miR-23b nanoparticles did not possess the efficient anti-inflammatory ability under our experimental conditions. As required, we have cited the reference and discussed it in both Introduction and Discussion sections. Please refer to line 102-106 in page 5 and 584-590 in page 25 for details.

Reviewer #3 (Remarks to the Author):

This manuscript describes the modification of 5th generation PANAM dendrimers which contain 128 terminal amines by perfluorobutyric anhydride. They claim the reaction modified approximately 48 of the 128 terminal amines and makes the modified dendrimer more hydrophobic.

They form a dendriplex with plasmid DNA that encodes the sequence for miR-23b. They characterize the complex and use it to transfect cells in culture and organs in vivo. They use a rat arthritis model and show data that indicates when the miR-23b plasmid nanoparticles were intravenously injected into the adjuvant-induced arthritis (AIA) rat model the RA symptoms with respect to joint swelling were eliminated and there was reduced bone damage and the rats recovered mobility.

Response: Thank you for your valuable comments that help us to improve our manuscript. We understood and agreed with the reviewer's concern regarding the appropriate control groups in our study. To address the issues, the corresponding experiments have been performed, and the new data have been offered in the revised manuscript. We wish that the thorough revision makes it more convinced and readable.

C 1: Unfortunately, the authors do not appear to have used the correct control and hence are not able to conclude that the fluorinated dendrimer was more active than the unfluorinated PANAM 5th generation dendrimer because they did not use the butyric acid

modified dendrimer in their experiments.

They used the unmodified 5th generation dendrimer as the control to deliver the miR-23b plasmid. I base my observation on supplemental figure 4 which indicates the zeta potential of the N/P 2/1 complex is +29 mV for the PANAM dendrimer but only +23 for the Fluorinated dendrimer. If the PAMAM dendrimer had been butylated the 2/1 N/P complex should have had a similar zeta potential as the 2/1 fluorinated dendrimer plasmid DNA complex.

C 5: The authors need to clarify these issues and if the appropriate control dendrimer that is the butylated 5th generation has not been used for the comparisons in the data this must be done.

Otherwise one cannot know if fluorinating the dendrimer has any beneficial effect on gene transfer compared to the butylated dendrimer.

R 1 and 5: Thank you for your constructive suggestion. Because your comments 1 and 5 involve the same issue regarding the butylated PAMAM as the appropriate control, we sought to address them together. Indeed, we agree with reviewer's comments that using butylated dendrimers as the control was more convinced to elucidate whether and why fluorinated dendrimers were more active than the butylated ones. Thus, the comparisons between fluorinated PAMAM and butylated ones have been performed to validate our conclusions. We synthesized the butylated PAMAM (termed as HP) by using the butyric anhydride and chemically characterized it by ^1H NMR (**Supplementary Fig. 4**). According to the ^1H NMR spectrum, an average number of 28 butyric acid groups were conjugated to each G5 PAMAM dendrimer, whose conjugation extent was close to the fluorinated PAMAM (FP) (29 fluorine groups modified on the surface of PAMAM). As required, we repeated the zeta potential measurement of PAMAM/miR-23b, FP/miR-23b and HP/miR-23b nanoparticles, respectively. As shown in **Supplementary Fig. 6a**, the FP/miR-23b and HP/miR-23b nanoparticles shared similar zeta potential, lower than PAMAM/miR-23b nanoparticles at the identical N/P ratios. This result indicated that the decreased surface charge of dendrimers was strongly dependent on the modification extent of PAMAM, where the loss of the primary amine groups would reduce the charge of PAMAM derivatives. Moreover, as shown in **Fig. 1** and **Supplementary Fig. 5**, HP/miR-23b and FP/miR-23b nanoparticles demonstrated similar DNA binding ability, where the plasmid was retarded at a similar N/P ratio of 2.0 by HP and FP. In contrast, the PAMAM/miR-23b nanoparticles started the complexion at the N/P ratio of 0.5 and realized the complete retardation near N/P ratio of 2.0, suggesting that PAMAM/miR-23b nanoparticles were more efficient in the binding with plasmids than HP and FP due to the higher charge density of PAMAM. Compared with the unmodified PAMAM, the transfection efficiency of HP significantly decreased at the identical N/P ratios in different cell lines, as evidenced by the observed fluorescence signal in **Fig. 2a, b** and **Supplementary Fig. 8**). This phenomenon indicated that the modification of butyric acid on the surface of PAMAM reduced its positive charge density and further inhibited the cellular uptake, leading to the decreased transfection efficiency. More intriguingly, we found that the fluorinated PAMAM achieved higher transfection efficiency than PAMAM at the identical N/P ratios in different cell lines and exhibited comparable transfection

ability with the commercial transfection reagent Lipofectamine 2000, thus convincingly validating that the fluorination improved the transfection efficacy of dendrimers. Normally, the transfection efficiency of FP was supposed to be reduced due to the decreased charge density after the modification; thus, we verified the role of the fluorine on the transfection efficiency. As shown in **Supplementary Fig. 9** and **Fig. 2c**, FP/miR-23b nanoparticles could be internalized by the cells *via* the clathrin-dependent endocytosis and escape from endosomes after 4 h while most of PAMAM/miR-23b nanoparticles were entrapped in the acidic vesicles, indicating that the fluorination could improve the endosomal escape of nanoparticles. Moreover, the fluorination strategies have been widely applied in other cationic polymers such as polyethylenimine and poly(propylenimine) which endowed these polymers with both hydrophobic and lipophobic properties to facilitate the cellular uptake and endosomal escape of nanoparticles, thereby achieving the enhanced gene transfection (Wang, et al. *Nat. Commun.* 2014, 5: 4053; Liu, et al. *Biomaterials* 2014, 35: 5407; Wang, et al. *Biomaterials* 2014, 24: 6603). Taken together, we compared the butylated PAMAM and fluorinated ones, and confirmed that the carrier FP possessed superior transfection efficiency, suggesting that the fluorine modified on the surface of PAMAM dendrimer played an essential role in the gene transfection. Please refer to Fig. 1 and 2, Supplementary Fig. 4-6 and 8-9, and the highlighted version of Results (page 133-157, 160-163, line 165-182 and line 196-200 in page 7-9) and Discussion (line 541-545 in page 23).

C 2: Line 119, figure 1 b. The authors state that the "To verify the binding and condensation ability of FP with oligonucleotide, the derivative FP and miR-23b were incubated together at different N/P ratios and the formed nanocomplexes were evaluated using gel retardation assay"

Is this an oligonucleotide or a plasmid DNA? The figure needs a molecular weight scale on it. If the authors used both oligonucleotides and plasmid DNA in this manuscript, they should state this in the methods and in the results where relevant. As it stands it is confusing to the reader.

R 2: Thank you for your careful check. We sincerely apologize for the vague description. The plasmid expressing miR-23b was used instead of oligonucleotide in the present study. Relevant description has been added in the Methods as well as the Results sections in a detailed manner. Moreover, we repeated the gel retardation assay to confirm the DNA binding and condensation ability of PAMAM/miR-23b, HP/miR-23b and FP/miR-23b nanoparticles and the DNA ladder of 10,000 bp was used as the molecular weight scale to avoid the confusion. Please refer to Fig. 1, Supplementary Fig. 5 and the highlighted version in Results (line 138-141 in page 7) and Methods (line 669-674 in page 29).

C 3: Depending upon the core of the dendrimer, the 5th generation PANAM has a molecular weight = 28,800, if they attached 48 perfluorobutyric acids to the PANAM the MW should have been approximately 38,304. In supplemental Figure 2 they state their

modified dendrimer has a molecular weight = 34,874, thus the number of amines modified would be closer to 31 not 48 as stated in sup Figure 2.

The authors should run a MALDI on the starting PANAM dendrimer so they can obtain a better estimate of the extent of modification. They should also estimate the number of fluorine groups on the dendrimer from the fluorine NMR of their modified dendrimer.

R 3: Thank you for your careful check and kind suggestion. As required, we performed the MALDI-TOF MS analysis of the initial PAMAM dendrimer. As shown in **Supplementary Fig. 3**, the molecular mass of G5 PAMAM dendrimer was determined to be 28859, which was rather close to the theoretical molecular weight of 28826 based on the ideal structure. According to the molecular weight of synthesized FP determined by MALDI-TOF MS, the number of heptafluorobutyric acid modified on each G5 PAMAM was calculated to be approximately 30. Moreover, we are grateful to the reviewer for the suggestion regarding the determination of fluorine content by ^{19}F NMR. Indeed, ^{19}F NMR has a broad chemical shift range and avoids the signal overlap, which makes it uniquely suitable for the quantification of fluorine-containing compounds. However, trifluoroacetic acid (TFA), used as the internal reference in ^{19}F NMR analysis is harmful and possibly caused severe skin burns when it is inhaled at a low concentration. Thus, considering the safety concern, we used an alternative to address the reviewer's question by a well-established ninhydrin assay (Wang, et al. *Nat. Commun.* 2014, 5: 4053). The calibration curve was established according to the dendrimer concentration and the absorbance (**Supplementary Fig. 2**). The number of fluorinated groups modified on each G5 PAMAM dendrimer were determined by the calibration curve ($Abs = 0.5986x - 0.0475$, $R^2 = 0.9907$, Abs represented the absorbance of sample at 570 nm, x was the concentration of primary amine groups, nmol). As the consequence, the remaining number of primary amine groups was calculated to be 98.75, indicating that approximately 29 of primary amine groups were modified by heptafluorobutyric acid. This result was consistent with the estimation by MALDI-TOF MS analysis. Thus, the reviewer's comment is correct, and we are sincerely appreciated for pointing out this mistake. Please refer to Supplementary Fig. 2 and Fig. 3, Supplementary methods and the highlighted version in Results (line 123-133 in page 6-7) and Methods (line 660-662 in page 29).

C 4: In addition, the 5th generation dendrimers used in these experiments are not optimized for gene delivery. The most active dendrimers have carboxylate defects in them. These defects significantly enhance their gene transfer active as reported in: Tang et al. *Bioconjug Chem* Nov-Dec 1996;7(6):703-14. doi: 10.1021/bc9600630.

R 4: We appreciate for your valuable comments and providing us with an in-depth study on the transfection of PAMAM dendrimer. PAMAM dendrimers are well-defined, highly branched, and nanoscale macromolecules with numerous amine groups on the surface, which have been widely used as non-viral vectors for the gene delivery due to their efficient transfection and limited immunogenicity (Luo, et al. *Acta. Biomater.* 2016, 43: 14). Particularly, considering the effective transfection, desirable biocompatibility and conformational flexibility, the G5 PAMAM dendrimer has been widely adopted for the delivery of plasmid DNA in gene therapy (Wang, et al. *Nat. Commun.* 2014, 5: 4053,

Kretzmann, et al. *Chem. Sci.* 2017, 8: 2923, Han, et al. *Biomater. Sci.* 2017, 5: 2268, Wang, et al. *Acta Biomater.* 2016, 46: 204). However, the study suggested by the reviewer indicated that the degradation of PAMAM not only lost the primary amines but generated the carboxylic acid groups upon the hydrolysis at the amide bonds. The carboxylate residues may form the zwitterion pairs with nearby tertiary amines, leading to the decreased positive charge. That's true, but on the other hand, this random process gave a mixed population of dendrimer molecules slightly differing in molecular weight and structure, and also led to these dendrimers with higher degree of flexibility (Dennig, et al. *J. Biotechnol.* 2002, 90: 339). The more flexible structure of G5 PAMAM was beneficial for the transfection of DNA (Navarro, et al. *Nanomedicine* 2009, 5: 287). The reference provided by the reviewer concluded that the key factors influencing the transfection of PAMAM dendrimer included: (1) high positive charge at physiological pH, (2) the presence of groups facilitating the endosome escape and (3) a highly flexible structure which can collapse and swell. Thus, how to manipulate these three properties is priority for the efficient transfection of PAMAM and its derivatives. The decreased charge density of degraded PAMAM will lead to the reduced electrostatic interaction between PAMAM and DNA or cell membrane which might influence the transfection efficiency. To address this issue, one alternative strategy should be considered to introduce other types of interactions or characteristics by modifying PAMAM dendrimer to acquire efficient transfection and low cytotoxicity, in which fluorinated strategy was exactly applied to balance three properties mentioned above to achieve enhanced transfection efficiency. Although the G5 PAMAM dendrimer partially lost the positive charge density after the fluorinated modification, the improved hydrophobic and lipophobic properties in turn increased the cellular uptake and endosome escape ability, leading to improved transfection efficacy (**Fig. 2** and **Supplementary Fig. 8**). Taken together, as you correctly pointed out, we agree and believe that the G5 PAMAM dendrimer may be not the best option for the gene delivery, but the fluorination strategy endowed the dendrimer with better transfection efficacy in comparison to the unmodified ones. We have taken this into account and revised the manuscript to make this point clear. Please refer to the response to C1 and the highlighted version in Introduction section (line 93-95 in page 5), Results section (line 168-178 and 196-200 in page 8-9) and Discussion section (line 541-545 in page 24).

Reviewer #4 (Remarks to the Author):

This manuscript present data concerning the development of FP/miR23b which was employed to intravenously treat AIA. While clinically FP/miR23b appeared to be effective, there are a number of concerns with the manuscript.

Response: We appreciate your contribution to reviewing and improving the manuscript. To address the issues raised, we performed additional experiments on mice bone marrow-derived macrophages (BMDMs) and elucidated the *in vivo* mechanism by using both adjuvant-induced arthritis (AIA) rats and collagen-induced arthritis (CIA) mice as models. Meanwhile, the statistical analysis has been

presented appropriately. Please find our itemized response below. We hope that the current revision makes it more convinced which can fulfill the standards of the journal.

C 1: It is unclear how the *in vitro* transfection experiments relate to the *in vivo* effect. LPS-stimulated RAW cells are very different from the inflammation of AIA *in vivo*.

R 1: We are grateful for your valuable comments. Here we used the LPS-stimulated RAW264.7 cells for two reasons. **First**, the RAW264.7 cells were used to evaluate the transfection efficiency and mechanism of FP/miR-23b nanoparticles. We concluded that the fluorination strategy improved the miR-23b transfection by the enhanced endosomal escape (**Fig. 2**). Accordingly, this finding also explained why the fluorinated PAMAM-mediated miR-23b delivery exhibited stronger therapeutic effect than the unmodified PAMAM in arthritic models. **Second**, these cells were adopted to reveal whether miR-23b regulated the anti-inflammatory process by targeting TAB2, TAB3 and IKK- α . Previous study has demonstrated that the downregulation of miR-23b played an essential role in the development of many autoimmune diseases, which can target TAB2, TAB3 and IKK- α in the NF- κ B signaling pathway (Zhu, et al. *Nat. Med.* 2012, 18: 1077). Meanwhile, LPS could stimulate RAW264.7 macrophages to generate pro-inflammatory cytokines such as IL-6, TNF- α and IL-1 β by the activation of NF- κ B signaling pathway through upregulating the expression level of TAB2 and TAB3 (Li, et al. *Int. Immunopharmacol.* 2017, 45: 110). Thus, these cells were usually adopted to evaluate the anti-inflammatory effect in many inflammation-related diseases (Simion, et al. *Nat. Commun.* 2020, 11: 6135; Sheedy, et al, *Nat. Immunol.* 2009, 11: 141; Liang, et al. *Nat. Commun.* 2018, 9: 4291). Further, we found that the expression of intracellular miR-23b decreased along with the pro-inflammatory cytokines' generation when the RAW264.7 cells were treated with different concentrations of LPS, suggesting that miR-23b played an inflammatory regulator in the macrophages (**Supplementary Fig. 10**). Moreover, the LPS stimulation increased the intracellular TAB2, TAB3 and IKK- α levels to activate the NF- κ B signaling pathway (**Supplementary Fig. 14**). Thus, the LPS-stimulated RAW264.7 cells could be an ideal cell model to investigate the anti-inflammatory response of FP/miR-23b nanoparticles. Please refer to Supplementary Fig. 10, Supplementary Fig. 14 and the highlighted version in Results (line 210-214 in page 10 and line 281-285 in page 13) and Discussion (line 547-550 in page 24). In addition, to strengthen the conclusion of the study, we followed the reviewers' valuable suggestions regarding the utilization of other cell lines and isolated the bone marrow-derived macrophages (BMDMs) to evaluate the biological function of FP/miR-23b. The studies using BMDMs also showed the similar pattern to RAW264.7 cells. Please refer to the response to C2 in details.

More importantly, we agree with the reviewer's comments and confess that the LPS-stimulated RAW264.7 cells or BMDMs are very different from the inflammation of AIA rats *in vivo* since the pathology of AIA is much more complicated. Thus, to confirm the therapeutic effect and elucidate the underlying mechanism of FP/miR-23b nanoparticles, we conducted more *in vivo* studies by using two different arthritic models including AIA

rats and collagen-induced arthritis (CIA) mice. Please refer to the response to C3 in details.

C 2: what are the effects of FP/miR23b on primary macrophages (in vitro developed from bone marrow or from human peripheral blood monocytes)?

R 2: We are grateful for your valuable comments. Due to the consequence of COVID-19, people were not allowed to make more trips outside, so that we could not easily obtain the human peripheral blood monocytes isolated from healthy donors. As an alternative, we isolated the bone marrow-derived macrophages (BMDMs) from mice to perform the *in vitro* experiments (Deng, et al. *Nat. Commun.* 2021, 12: 2174; Mayes-Hopfinger, et al. *Nat. Commun.* 2021, 12: 4546; Kobayashi, et al. *Nat. Commun.* 2016, 7: 11624). When BMDMs were treated with FP/miR-23b nanoparticles, we found that the nanoparticles effectively inhibit the expression of IL-1 β , IL-6 and TNF- α , indicating that miR-23b achieved an obvious anti-inflammatory effect in macrophages (**Fig. 4a-f**). Further, qPCR analysis and Western blotting revealed that the FP-mediated miR-23b delivery effectively decreased the intracellular expression of IKK- α , TAB2 and TAB3 in BMDMs, indicating that miR-23b manipulated the NF- κ B signaling pathway to achieve the anti-inflammatory effect (**Fig. 4g-j**). Besides the inhibition of inflammation, we identified whether miR-23b could regulate the cell proliferation of BMDMs, since the proliferation of macrophages is still abundant in the synovium of RA. As shown in **Fig. 3 and Supplementary Fig. 11**, the apoptosis of BMDMs was triggered after the cells were treated with FP/miR-23b nanoparticles, suggesting that miR-23b regulated the cell apoptosis to achieve the anti-proliferative function of macrophages. Thus, we are convinced by the conclusion that miR-23b could trigger the cell apoptosis to achieve the inhibition of cell proliferation. Integrating the consistent results obtained in RAW264.7 cells and BMDMs, we concluded that miR-23b could not only modulate the inflammation but also inhibit the cell proliferation, showing the therapeutic potential of miR-23b in experimental rheumatoid arthritis. The experiments, results and appropriate discussions achieved in BMDMs have been supplemented in the revised manuscript and Supplementary information, and the original results of biological effects acquired in RAW264.7 cells have been presented in Supplementary information. Please refer to Fig. 3 and 4, Supplementary Fig. 11-15 and the highlighted version in Results (line 225-263, line 267-276 and line 281-290 in page 11-13), Discussion (line 547-553 in page 24) and Methods (line 675-681 in page 29-30 and line 751-755 in page 34).

C 3: many of the *in vitro* effects are artificial (cell lines and transfection and LPS), making it unclear what the mechanism for effectiveness *in vivo* is.

R 3: We agree with reviewer's point, and it is essential to elucidate the underlying mechanism by which the FP/miR-23b nanoparticles realized the *in vivo* therapeutic effect in the experimental RA. To address this issue, extensive efforts have been made to confirm the therapeutic efficacy of FP/miR-23b nanoparticles in the experimental RA and clarify the detailed mechanism behind (AIA rats and CIA mice). After the intravenous injection of FP/miR-23b nanoparticles into the arthritic rats and mice models, the nanoparticles could be favorably accumulated in the arthritic joints through the ELVIS

effect and distributed in the hyperplastic synovium (**Fig. 9**). The nanoparticles were captured by the synovial macrophages as well as the fibroblasts in a non-specific manner and further increased the miR-23b expression in the synovium (**Supplementary Fig. 25**). Previous study has revealed the anti-inflammatory role of miR-23b in synovial fibroblasts (Zhu, et al. *Nat. Med.* 2012, 18: 1077). The current study also confirmed that the delivered miR-23b could target TAB2, TAB3 and IKK- α to regulate the NF- κ B signaling pathway in the synovium (**Fig. 10a-d and Supplementary Fig. 26**). Besides, we showed that the FP/miR-23b nanoparticles could inhibit the cell proliferation in hyperplastic synovium by triggering the apoptotic effect (**Fig. 10e and Supplementary Fig. 27**). Previous reports have shown that the inhibition of NF- κ B signaling pathway could reduce the level of Bcl-2 family, leading to the loss of mitochondrial membrane potential, which was probably an important reason for the induction of cell apoptosis by miR-23b *via* the mitochondrial dependent apoptotic pathway (Pagliari, et al. *Mol. Cell Biol.* 2000, 20: 8855). As the consequence, the administration of FP/miR-23b nanoparticles decreased the pro-inflammatory cytokines secreted in the synovium and protected the bone and articular cartilage from the erosion. Moreover, the symptoms of RA have been mitigated as characterized by the reduced paw swelling and recovered athletic ability. Collectively, the blockage of NF- κ B signaling pathway by miR-23 broke the vicious loop of proliferation and inflammation in the arthritic joints to realize the potential therapeutic efficacy. We assumed that this synergistic effect could be a compelling strategy for RA treatment. Please refer to Fig. 9, Fig. 10, and Supplementary Fig. 25-27, and the highlighted version in Results (line 452-484 and line 486-513 in page 20-23), Discussion (line 591-596 in page 26) and Methods (line 852-881 in page 37-38).

C 4: a description of the rats ability to walk are more than is needed to get the point across, while quantitation of histology and mechanistic studies of what is going on in the joint are not presented.

R 4: We agree with the reviewer that understanding the mechanism of action will be essential and important in exploiting the findings of our study. As required, we have performed more experiments to elucidate the mechanism and summarized the results in **Fig. 9, Fig. 10** and **Supplementary Fig. 25-27**. Please refer to the response to C3 above. We repeated the experiments on AIA rats and CIA mice and performed the histological quantification of the arthritic joints after the H&E staining (**Supplementary Fig. 17 and Supplementary Fig. 21**). FP/miR-23b nanoparticles effectively inhibited the cellular infiltration and mitigated the histological features of synovitis, which was beneficial for the experimental RA treatment. Please refer to Supplementary Fig. 17 and Supplementary Fig. 21 and the highlighted version in Results (line 383-390 in page 17 and line 433-436 in page 19) and Methods (line 844-851 in page 37).

C 5: In figure 10, the authors provide a cartoon, suggesting the mechanism of action, but they present no *in vivo* data to support this mechanism, other than a reduction of cytokines. In fact, the data in Supp Fig 12, uses a t-test to show difference, but they mention ANOVA with a post-test, which is correct. What is "control" and why is FP/miR23b not different?

R 5: Thank you for your valuable suggestion. Accordingly, we have supplemented the studies on *in vivo* mechanism to support our conclusions. Please refer to the response to C3 above. With regard to discrepancy in animal experiments, the samples were collected from at least 3 different mice in each group. We evaluated the statistical difference between two groups, so unpaired Student's *t*-test was appropriate for the statistical analysis. In **Supplementary Fig. 25** (which was Supplementary Fig. 12 in the original manuscript), the control represented the healthy rat without arthritis. After the AIA rats were treated with FP/miR-23b nanoparticles, the miR-23b expression was restored to a healthy level, which was higher than the untreated AIA rats with statistical significance ($p = 0.0032$). We were sincerely sorry for our vague description, and it has been revised in a clear way. Please refer to Supplementary Fig. 25 and the highlighted version in Results (line 490-495 in page 22).

C 6: Specifically, does the FP/miR23b get into synovial macrophages or fibroblasts, is there increased apoptosis, is there an effect on NF-kb?

R 6: Thank you for your kind comments. In attempt to address this issue, we labelled the miR-23b plasmid with Cy5 dye and further tracked the fluorescence of FP/Cy5-miR-23b nanoparticles after the intravenous injection to CIA mice. As shown in **Fig. 9a-d**, FP/miR-23b nanoparticles were able to accumulate in the arthritic joints and still manifested the fluorescence until 24 h post-injection due to the Extravasation through Leaky Vasculature and subsequent Inflammatory cell-mediated Sequestration (abbreviated as ELVIS) in the arthritic mice, which was consistent with previous reports (Yuan, et al. *Adv. Drug Deliv. Rev.* 2012, 64: 1205; Wang, et al. *Biomater. Sci.* 2017, 5: 1407). Besides, the fluorescence of Cy5-miR-23b was uniformly distributed at the site of the hyperplastic synovium, which was probably attributed to the leakage of nanoparticles through the invasive pannus (**Fig. 9e**). The synovial macrophages and fibroblasts (FLSs) were visualized by the immunofluorescence of F4/80 and cadherin-11, respectively. As shown in **Fig. 9f**, the nanoparticles were observed to preferably accumulate in the synovial macrophages and FLSs, indicating that FP was non-specifically captured by the synoviocytes. Moreover, the accumulation of miR-23b significantly decreased the expression of TAB2, TAB3 and IKK- α in the hyperplastic synovium, indicating that the delivered miR-23b could regulate the NF- κ B signaling pathway in the experimental RA (**Fig. 10a-d**, **Supplementary Fig. 25**, and **Supplementary Fig. 26**). Besides, we evaluated the apoptotic effect in the synovium of arthritic models by TUNEL and Ki67 staining. As shown in **Fig. 10e and Supplementary Fig. 27**, as the consequence of miR-23b delivery, the synovium of arthritic animals showed the inhibition of cell proliferation due to the induction of apoptosis, as observed by the TUNEL signal in the FP/miR-23b group and PAMAM/miR-23b group. This result suggested that miR-23b triggered the apoptotic effect in the hyperplastic synovium of the animals bearing the arthritis. Collectively, we inferred that miR-23b realized the anti-inflammatory and anti-proliferative effects in the experimental RA treatment. Please refer to Fig. 9-10, Supplementary Fig. 25-27, and the highlighted version in Results (line 452-484 and line 486-513 in page 20-23), Discussion (line 591-596 in page 26) and Methods (line 852-881 in page 37-38).

C 7: it appears the AIA was only performed in a single experiment and this needs to be repeated.

R 7: We are grateful for your constructive compliments. As required, we have repeated the experiments on AIA rats and also performed the mechanistic evaluation of miR-23b by immunohistology and qPCR analysis. These results have been updated without significant changes and reached a similar conclusion. Moreover, to strengthen the manuscript, we utilized the CIA mice as the second and independent model to validate the *in vivo* therapeutic efficacy and mechanism of miR-23b.

In our repeated experiments of AIA rats, the FP/miR-23b nanoparticles still exhibited an anti-arthritic effect, as assessed by the reduced paw swelling and edema, decreased clinical indexes as well as the recovered athletic ability (**Fig. R4**). In contrast, the FP/NC nanoparticles did not show therapeutic efficacy. These data showed a similar phenomenon as described in our original manuscript, and thus we decided to keep the data from our original version without any changes (Fig. 6). Moreover, the miR-23b delivery inhibited the arthritic inflammation, protected the bone and cartilage erosion, and inhibited the infiltration of synoviocytes (**Fig. 7, Fig. 8** and **Supplementary Fig. 17**). These results were consistent with our initial study and confirmed the potential therapeutic efficacy in the treatment of experimental RA. To evaluate the biocompatibility of FP/miR-23b nanoparticles, we repeated the blood biochemical examination including ALT, AST, ALP, AST, BUN as well as CREA, and analyzed the histological change of major tissues (**Supplementary Fig. 18**). As expected, the FP/miR-23b nanoparticles exhibited no systemic toxicity. Based on these studies, we were inspired to elucidate the *in vivo* mechanism of FP/miR-23b nanoparticles. In addition, FP/miR-23b nanoparticles could be preferably located in the synovial macrophages and FLSs, which exerted the anti-inflammatory and anti-proliferative responses by targeting TAB2, TAB3 and IKK- α in the NF- κ B signaling pathway (**Fig. 9 and 10**). Please refer to Fig. R4, Fig. 7, Fig. 8, Fig. 10, Supplementary Fig. 17 and Supplementary Fig. 18 for details.

Editorial note: Figure redacted

C 8: this model should not be referred to as RA, it is an experimental model of RA.

R 8: We are very appreciated for your kind comments. As you correctly pointed out, we have revised the correlated items to experimental rheumatoid arthritis, which made the manuscript more precise (e.g. title, line 107-108 in page 6, line 448-450 in page 20, line 510-513 in page 22-23 and line 604-606 in page 26).

C 9: ELVIS is referred to in the abstract with no explanation of what this refers to until the introduction. If the authors want to suggest that their mechanism is working through this mechanism, they should define the parameters that permit them to make this claim.

R 9: According to your insightful suggestion, we have revised the abstract and performed additional experiments to make this point clear. Recent studies found that the nanostructure particles could passively get access to the inflammatory sites via a new retention mechanism termed as ELVIS (extravasation through leaky vasculature and subsequent inflammatory cell-mediated sequestration), a process analogous to the classical enhanced permeability and retention (EPR) that allows the nanoparticles to be accumulated in the solid tumors (Metselaar, et al. *Ann. Rheum. Dis.* 2004, 64: 348; Yuan et al. *Adv. Drug Deliv. Rev.* 2012, 12: 1205). The widely used nanoparticles such as micelles, liposomes and polymers have been demonstrated to achieve the targeted delivery to the inflamed sites *via* ELVIS effect. Thus, we were wondering whether the FP/miR-23b nanoparticles could be preferably accumulated in the inflamed joints. In an attempt to address this issue, we labelled the miR-23b plasmid with Cy5 dye and tracked the fluorescence distribution of the FP/Cy5-labelled miR-23b nanoparticles in CIA mice after the intravenous injection. As shown in **Fig. 9a-d**, the FP/Cy5-labelled miR-23b nanoparticles showed the fluorescence signal in the arthritic joints as early as 1 h after the administration, and the fluorescence still persisted until 24 h. In contrast, the fluorescence was diminished in the limbs of CIA mice treated with free Cy5-labelled miR-23b at 24 h. These results demonstrated the carrier FP could prolong the circulation of miR-23b and achieve the selective accumulation of nanoparticles in the inflamed tissues due to the ELVIS effect. Please refer to Fig. 9a-d, and the highlighted version in Abstract (line 23-25 in page 2), Results (line 452-468 in page 20-21) and Discussion (line 591-596 in page 26).

REVIEWER COMMENTS

Reviewer #1 (Remarks to the Author):

My comments have been adequately addressed, and this manuscript presents an interesting new delivery method to enhance miRNA therapeutics in mammals.

Reviewer #2 (Remarks to the Author):

The authors have responded appropriately to all of my outstanding concerns in their last revision!

As such, from my perspective I recommend publication in its current form.

Reviewer #3 (Remarks to the Author):

Fluorinated polyamidoamine dendrimer-mediated miR-23b delivery for the treatment of experimental rheumatoid arthritis

Han et al.

General: The authors have not provided a compelling rebuttal to the comments related to the synthesis and characterization of the fluorinated dendrimer used in the reported studies nor have they included the appropriate controls (HP-dendrimer) in most of the studies that are included in this manuscript.

Original submission methods

484 Synthesis and characterization of FP. Heptafluorobutyric anhydride (54.12 mg) and
485 PAMAM (50.00 mg) were dissolved in 8 ml methanol. The mixture was stirred at room
486 temperature for 48 h, and then dialyzed against distilled water for another 48 h (MWCO:
487 3500 Da). The product was obtained through lyophilization and subjected to the structural
488 characterization using ¹⁹F NMR on AVANCE III spectrometer (Bruker, Rheinstetten, 489
Germany). The molecular weight of FP was determined by AB SCIEX 5800 MALDI-TOF
490 mass spectrometer (Framingham, Massachusetts), in which 2,5-dihydroxybenzoic acid was
491 used as a matrix. The butyric acid-modified PAMAM was synthesized as described above
492 except for the usage of butyric anhydride (20.88 mg), and characterized using ¹H NMR on
493 AVANCE DMX 500 NMR spectrometer (Bruker, Rheinstetten, Germany).

Results original submission:

"According to the ¹⁹F NMR spectra (Supplementary Fig. 1), the characteristic
110 peaks of heptafluorobutyric anhydride could be clearly observed in FP, and the peak (a) in
111 heptafluorobutyric anhydride significantly shifted to the peak (a') in FP after the
112 conjugation, indicating that PAMAM was successfully modified with heptafluorobutyric
113 acid. Next, to determine the fluorination degree of FP, heptafluorobutyric acid was replaced
114 by butyric acid to obtain the butyric acid-modified PAMAM using an identical synthesis
115 route, and an average number of 48 butyric acids were calculated to be conjugated to each
116 PAMAM molecule (Supplementary Fig. 2). Accordingly, the molecular weight of FP was
117 predicted to be 34874.0 Da. Using MALDI-TOF MS analysis, the molecular weight of FP
118 was determined to be 35017.4 Da,"

Revised manuscript methods

line 655 Synthesis and characterization of FP. Heptafluorobutyric anhydride (54.12 mg) and PAMAM (50.00 mg) were dissolved in 8 ml methanol. The mixture was stirred at room temperature for 48 h, and then dialyzed against distilled water for another 48 h (MWCO: 3500 Da). The product was obtained through lyophilization and subjected to the structural characterization using ¹⁹F NMR on AVANCE III spectrometer (Bruker, Rheinstetten, Germany). The molecular weight values of PAMAM and FP were determined by AB SCIEX 5800 MALDI-TOF mass spectrometer (Framingham, Massachusetts), in which 2,5-dihydroxybenzoic acid was used as a matrix. The butyric acid-modified PAMAM was synthesized as described above except for the usage of butyric anhydride (20.88 mg), and characterized using ¹H NMR on AVANCE DMX 500 NMR spectrometer (Bruker, Rheinstetten, Germany).

Results revised manuscript results

line 114 Synthesis and characterization of fluorinated PAMAM. In recent years, dendrimer PAMAM has been widely utilized in gene delivery, but its transfection efficiency was still limited especially in synoviocytes⁴⁵. To address this issue, a facile fluorination method was adopted to improve its transfection efficiency, in which the amine groups of PAMAM were modified with perfluoro acid anhydride to construct a fluorinated derivative, namely FP (Fig. 1a). According to the ¹⁹F NMR spectra (Supplementary Fig. 1), the characteristic peaks of heptafluorobutyric anhydride could be clearly observed in FP, and the peak (a) in heptafluorobutyric anhydride significantly shifted to the peak (a') in FP after the conjugation, indicating that PAMAM was successfully modified with heptafluorobutyric acid. Next, to determine the fluorination degree of FP, the number of perfluoro acid on the surface of each FP dendrimer was determined by ninhydrin assay.³⁸ As shown in Supplementary Fig. 2, the amine groups available on the surface of FP were calculated to be 98.75, suggesting that approximately 29 of primary amine groups were modified by heptafluorobutyric acid. Moreover, MALDI-TOF MS analysis was conducted to confirm the molecular weight of unmodified PAMAM and FP. As shown in Supplementary Fig. 3, a general molecular weight distribution was well observed, where the molecular weight values of PAMAM and FP were calculated to be 28859.7 and 35017.4, respectively. Therefore, the theoretical number of heptafluorobutyric acid modified on the surface of FP were determined to be approximately 30, which was consistent with the results obtained by ninhydrin assay. To verify the effect of chemical modification of PAMAM on the performance of DNA binding and condensation, the heptafluorobutyric acids were replaced by butyric acids to obtain the butyric acid-modified PAMAM (termed as HP) using an identical synthesis route. As shown in Supplementary Fig. 4, HP achieved an average number of 28 butyric acids conjugated to each PAMAM molecule, whose conjugation extent was close to FP.

There is no explanation for why the same synthetic reaction conditions used in the original version and revised versions of the manuscript resulted in modified dendrimers with differing molecule weights and extents of modification in the revised version compared (approximately 30) to the original submitted version⁽⁴⁸⁾. This is a serious flaw in the manuscript

Specific comments:

Line 100 The fluorinated dendrimer used in reference 38 was a fluorobenzoic acid modified dendrimer which is substantially different than the one used in the current manuscript.

line 217 free miR-23b and FP/mock miRNA (FP/NC) nanoparticles did not alter the miR-23b

What do you mean by this? a plasmid encoding a defective miR-23b sequence?

line 432 In contrast, the FP-mediated delivery of mock miRNAs exhibited no therapeutic efficacy.

What is the mock miRNAs see above comment ?

line 657 "The mixture was stirred at room temperature for 48 h, and then dialyzed against distilled water for another 48 h (MWCO: 3500 Da). The product was obtained through lyophilization and subjected to the structural"

This method of using distilled water to remove a carboxylate containing side product from the modified dendrimer may not remove all of the side product do a strong ionic interaction between and amines in the dendrimer and the carboxylates in the fluorobutyric acid of butyric acid

line 614_methods What are the sequence of the miR-23b containing plasmid and the " mock miRNAs (NC) " ?

There should be a plasmid map in the supplemental information Plasmid maps of both indicating enhancers/promoters/gene construct/etc should be included in the supplemental information.

It seems that statistical comparisons between the fluorinated and butylated dendrimers were not done in supplemental figures 6a or 6b

line 525. "Previously, Zhu et al. demonstrated that miR-23b could suppress the inflammatory response in RA. " -

This manuscript confirms that report.

line 584 Previous study reported that the dendrimer PAMAM alone presented the inherent anti-inflammatory ability by suppressing the COX activity,⁴⁴ whereas the FP-mediated delivery of mock miRNAs barely showed the therapeutic efficacy in the experimental RA treatment. We assumed that the anti-inflammatory effect of FP was diminished after the fluorinated modification.

What would be the mechanism for this, if only 30 out of 128 amines are modified? is this effect caused by a single dendrimer?

Reviewer #4 (Remarks to the Author):

The authors have responded robustly to my concerns. Two issues remain to be addressed.

1) throughout the manuscript are figures in which multiple comparisons are made and unpaired student's t-test applied. In this situation ANOVA with a post hoc analysis should be used.

2) the data demonstrates massive accumulation of the therapeutic compound in the liver, but there is not damage, at least by increase of enzymes that are released with liver damage. Have the authors looked at the liver for effects on other parameters as they did in the joint tissue? NF- κ B, apoptosis etc? Assuming there is no effect on the liver, this should at lease be discussed in the discussion.

Response to the comments from reviewers:

All authors would like to express our gratitude to the reviewers for their valuable and constructive comments. The aftermath of Covid still lingers. Due to the Omicron pandemic, we are ordered to stay and work at home, and the laboratories and animal facility have been closed since March 10, 2022. Thus, we were compelled to postpone our proposed animal experiments until August 3, 2022 and resumed the study afterwards. We sincerely apologized and would like to petition the reviewers for understanding the delay. Each comment raised by the reviewers have been taken into careful and sufficient account, and our response to each comment was provided below.

Reviewer #1 (Remarks to the Author):

My comments have been adequately addressed, and this manuscript presents an interesting new delivery method to enhance miRNA therapeutics in mammals.

Response: We greatly thank the reviewer for all the comments and suggested edits that contributed to improve the quality of the manuscript.

Reviewer #2 (Remarks to the Author):

The authors have responded appropriately to all of my outstanding concerns in their last revision! As such, from my perspective I recommend publication in its current form.

Response: Thank you very much for this positive assessment of our study and also for recommending our manuscript for publication.

Reviewer #3 (Remarks to the Author):

General: The authors have not provided a compelling rebuttal to the comments related to the synthesis and characterization of the fluorinated dendrimer used in the reported studies nor have they included the appropriate controls (HP-dendrimer) in most of the studies that are included in this manuscript.

Response: We thank the reviewer for bringing our attention to these points, which indeed greatly improve the quality of our study. Unfortunately, when we received the reviewer's comments, we were no longer able to perform any *in vitro* or *in vivo* experiment since we have been quarantined and worked at home for three months due to the latest COVID-19 pandemic. In response to the emergence of Omicron variants, the local government ordered the immediate and complete shutdown, which also extended to the labs and the animal facility in our university. Thus, we have no choice but to postpone the proposed animal studies. Recently, we resumed work and thus additional animal experiments could be performed as requested by the reviewer. Because reduction of experimental animals is an important principle for animal experiments, we minimized the number of animals by setting four groups to verify the therapeutic efficacy of HP-dendrimer, including normal group (healthy animals), untreated group (arthritic animals without treatment), FP/miR-23b group and HP/miR-23b group (arthritic animals treated with FP/miR-23b and HP/miR-23b nanoparticles, respectively). As shown in **Supplementary Fig. 25** and **Supplementary Fig. 26**, the treatment of HP/miR-23b nanoparticles did not alleviate the arthritic symptoms, as characterized by the severe paw swelling, damaged bone tissues, and irreversible cartilage erosion. Moreover, the HP-mediated miR-23b delivery did not reduce the level of pro-inflammatory cytokines. These results demonstrated that the HP/miR-23b nanoparticles did not possess therapeutic outcomes. To further investigate the therapeutic difference of FP/miR-23b and HP/miR-23b nanoparticles, we determined the miR-23b expression in the synovial tissues after the administration of these two nanoparticles. As shown in **Supplementary Fig. 27**, compared with untreated groups, the administration of HP/miR-23b nanoparticles did not significantly improve the synovial

miR-23b expression. As assumed, this phenomenon was mainly caused by the poor transfection efficiency of HP/miR-23b nanoparticles. Taken together, it was evident that the FP/miR-23b nanoparticles exhibited superior therapeutic efficacy than PAMAM/miR-23b and HP/miR-23b nanoparticles. Please refer to Supplementary Fig. 25-27 and the highlighted version in the Results section (line 448-462 in page 20).

Regarding the synthesis and characterization of the fluorinated dendrimer, we have carefully considered the related comments and discussed them in a point-to-point manner. Again, we really appreciate your efforts in reviewing our manuscript during this unprecedented and challenging time. Your careful review has helped us to make our study clearer and more convinced.

Comment (C) 1: There is no explanation for why the same synthetic reaction conditions used in the original version and revised versions of the manuscript resulted in modified dendrimers with differing molecule weights and extents of modification in the revised version compared (approximately 30) to the original submitted version (48). This is a serious flaw in the manuscript.

Response (R) 1: Thank you for your careful check. Indeed, we mistakenly calculated the fluorinated content of the derivative FP in the original version. We sincerely apologized for the mistake that confused the reviewer. In the revised manuscript, we have followed the reviewer's comments to determine the fluorinated content by MALDI-TOF MS analysis. As shown in Supplementary Fig. 3, the molecular weight of unmodified PAMAM and FP were determined to be 28859.7 and 35017.4, respectively, indicating that the number of the heptafluorobutyric acid modified on each G5 PAMAM was approximately 30. Moreover, the ninhydrin assay was conducted to determine the remaining amine groups on FP, in which approximately 29 of primary amine groups were modified by heptafluorobutyric acid (Supplementary Fig. 2). The error has been amended in our previous version of revised manuscript.

C 2: Line 100 The fluorinated dendrimer used in reference 38 was a fluorobenzoic acid modified dendrimer which is substantially different than the one used in the current manuscript.

R 2: Thank you for your comment. The fluorinated approach has been proven to improve the transfection efficiency of dendrimer such as PAMAM. In the study performed by Wang et al., the fluorinated PAMAM was constructed by using perfluoro acid anhydrides, in which heptafluorobutyric acid-modified dendrimers (termed as G5-F7) showed much higher transfection efficacy in HEK293 cells than other fluorinated dendrimers (**Fig. R1**). Accordingly, heptafluorobutyric anhydride was chosen as fluorinated reagent to construct the derivative FP in our study. Please refer to the Methods section in Ref. 38 (Wang, et al. *Nat. Commun.* 2014, 5: 4053). Meanwhile, we totally agreed with the reviewer that the modification of PAMAM dendrimer using fluorobenzoic acid was another facile approach of fluorination to improve the transfection efficiency of cationic dendrimers, which has already been proved (Wang, et al. *Biomaterials* 2014, 35: 6603-6613). Intriguingly, the modification with fluorobenzoic acid endowed PAMAM with higher cellular uptake ability, more efficient endosomal escape and faster intracellular DNA release, whose behaviors were similar to that of heptafluorobutyric acid-modified dendrimers. We have concluded this study in our revised manuscript. Please refer to ref. 47 in our revised manuscript.

Editorial note: Figure redacted

C 3: Line 217 free miR-23b and FP/mock miRNA (FP/NC) nanoparticles did not alter the miR-23b. What do you mean by this? a plasmid encoding a defective miR-23b sequence?

C4: line 432 In contrast, the FP-mediated delivery of mock miRNAs exhibited no therapeutic efficacy. What is the mock miRNAs see above comment?

R 3 and 4: We apologize for the confusion. In fact, the plasmid expressing the negative control miRNA (termed as NC miRNA) was constructed by GenePharma, in which the NC miRNAs are validated random sequences and have shown to produce no identifiable effects on known miRNA function. Thus, when FP/NC nanoparticles were transfected into the macrophages, the NC miRNAs did not alter the intracellular miR-23b expression. Additionally, the intravenous injection of FP/NC nanoparticles to arthritic animals did

not exhibit the therapeutic outcomes because NC miRNAs preserved no biological function. To avoid the confusion, we amended the words “mock miRNA” to “negative control (NC) miRNA” throughout the manuscript and also provided its plasmid map and sequence (**Supplementary Fig. 33**). Please refer to Supplementary Fig. 33 and the highlighted version in Results (line 217, 312 and 433) and Methods (line 637-640 in page 28).

C 5: line 657 "The mixture was stirred at room temperature for 48 h, and then dialyzed against distilled water for another 48 h (MWCO: 3500 Da). The product was obtained through lyophilization and subjected to the structural"

This method of using distilled water to remove a carboxylate containing side product from the modified dendrimer may not remove all of the side product do a strong ionic interaction between and amines in the dendrimer and the carboxylates in the fluorobutyric acid of butyric acid.

R 5: We thank the reviewer for raising this insightful comment. We agree with the reviewer that the residual heptafluorobutyric acid was partially adsorbed on the dendrimer due to the ionic interaction after dialysis. To improve the purity of the product, we have extended the dialysis process and used the alkaline phosphate buffer and distilled water as the dialysis buffer to remove the residual heptafluorobutyric acid. Moreover, although the existence of the adsorbed heptafluorobutyric acid might affect the plasmid binding and condensing process, the transfection efficacy of fluorinated dendrimer was still desirable, which did not change our conclusion. Again, we appreciate your kind suggestion and we will improve our synthetic strategy in future study based on your suggestion.

C 6: line 614_methods What are the sequence of the miR-23b containing plasmid and the " mock miRNAs (NC) "? There should be a plasmid map in the supplemental information

Plasmid maps of both indicating enhancers/promoters/gene construct/etc should be included in the supplemental information.

R 6: We thank the reviewer for the useful comment. As required, we have provided the plasmid map and sequence information of plasmids encoding miR-23b and NC miRNA. Moreover, miR-23b and NC miRNA were constructed to the same plasmid vector except for different miRNA sequences. Please refer to Supplementary Fig. 33 and the highlighted version in Methods (line 637-640 in page 28).

C 7: It seems that statistical comparisons between the fluorinated and butylated dendrimers were not done in supplemental figures 6a or 6b.

R 7: Thank you for your suggestion. Following your advice, we have performed the statistical analysis (comparisons) for the groups in **Supplementary Fig. 6**. For instance, when the N/P ratio reached 4.0, PAMAM/miR-23b nanoparticles showed higher zeta potential value compared with FP/miR-23b and HP/miR-23b nanoparticles with statistical significance (PAMAM/miR-23b vs. FP/miR-23b, $P=0.0155$; PAMAM/miR-23b vs. HP/miR-23b, $P=0.0065$; FP/miR-23b vs. HP/miR-23b, ns). In contrast, the hydrodynamic size of PAMAM/miR-23b nanoparticles was significantly smaller than HP/miR-23b and FP/miR-23b nanoparticles at the N/P ratio of 4.0 (PAMAM/miR-23b vs. FP/miR-23b, $P<0.0001$; PAMAM/miR-23b vs. HP/miR-23b, $P=0.0002$; FP/miR-23b vs. HP/miR-23b, ns). Please refer to Supplementary Fig. 6.

C 8: line 525. "Previously, Zhu et al. demonstrated that miR-23b could suppress the inflammatory response in RA. " This manuscript confirms that report.

R 8: We thank the reviewer for raising this comment. Indeed, in the study conducted by Zhu et al., miR-23b has been identified as the key regulator to maintain the inflammatory inhibition in fibroblast-like synoviocytes (FLSs) by targeting the NF- κ B signaling pathway, which was beneficial for RA treatment. We are aware of their previous work in the arthritis and have contributed to this literature. In the development of RA, besides

FLSs, the hyperplasia and infiltration of macrophages has also been identified as the main pathophysiological characteristics, leading to the inflammatory microenvironment. In our current study, we have confirmed that miR-23b was capable of targeting TAB2/3 and IKK- α to suppress the inflammatory response in macrophages using RAW264.7 cells and BMDMs as models. More importantly, besides known anti-inflammatory effect of miR-23b, we have also verified that the delivery of miR-23b efficiently inhibited the cell proliferation of macrophages by inducing the cell apoptosis, which was barely reported before to our best knowledge. The activated macrophage lining in RA synovium exhibited “tumor-like” features such as resistance to apoptosis, vigorous proliferation and increased invasion ability to bone and cartilage. Thus, it was of great significance that the miR-23b delivery could not only inhibit the inflammation but also induce the cell apoptosis, achieving the synergistic efficacy in RA treatment. Further, we confirmed the therapeutic efficacy of miR-23b delivery and revealed its *in vivo* mechanism using two experimental arthritis models, in which the miR-23b delivery was able to inhibit the inflammation and proliferation by targeting TAB2/3 and IKK- α in the synovium. Taken together, the miR-23b-based therapy substantially played dual roles of inflammatory inhibition as well as apoptosis induction and focused on breaking the vicious loop of proliferation and inflammation, aiming to achieve the enhanced therapeutic efficacy in the RA treatment. We believe that the current study provided a more direct investigation and a new perspective of miR-23b on experimental RA treatment.

C 9: line 584. Previous study reported that the dendrimer PAMAM alone presented the inherent anti-inflammatory ability by suppressing the COX activity,⁴⁴ whereas the FP-mediated delivery of mock miRNAs barely showed the therapeutic efficacy in the experimental RA treatment. We assumed that the anti-inflammatory effect of FP was diminished after the fluorinated modification. What would be the mechanism for this, if only 30 out of 128 amines are modified? is this effect caused by a single dendrimer?

R 9: We thank the reviewer for this valuable comment. Although the G4 PAMAM was reported to exhibit the inherent anti-inflammatory ability by suppressing the COX activity, we observed that the FP-mediated delivery of NC miRNA did not show any therapeutic efficacy. The reviewer is correct. Based on current experiments and data, we cannot easily conclude the anti-inflammatory effect of FP dendrimer alone. It is still unclear whether the fluorination could diminish or promote the anti-inflammatory effect. This mechanism is beyond our scope of current study and will be interrogated in ongoing and future studies. Thus, to make the conclusion more conscious, we decided to remove to the speculative part (line 584-590 in last revised version) from the discussion.

Reviewer #4 (Remarks to the Author):

The authors have responded robustly to my concerns. Two issues remain to be addressed.

Response: We would like to sincerely thank you for your insightful comments and detailed review of our manuscript. Unfortunately, due to the burst of COVID-19 variant Omicron in Changchun, the laboratories and animal facility in our university have been closed from March 10, 2022 to control the virus spread. Thus, we were compelled to postpone the proposed animal experiments until August 3 and resumed the study afterwards. We have carefully considered every comment and addressed them below. Again, we really appreciate your efforts in reviewing our manuscript during this difficult time. Your careful review has helped us to improve our manuscript clearer and more comprehensive.

C 1: throughout the manuscript are figures in which multiple comparisons are made and unpaired student's t-test applied. In this situation ANOVA with a post hoc analysis should be used.

R 1: We thank the reviewer for the careful check and agree that the ANOVA with a post hoc analysis should be performed in the situation of multiple comparisons. Thus, the statistical significance of different groups was calculated using one-way ANOVA with LSD test and the manuscript was finally reviewed by a statistician to assess the methods. Please refer to the figures in our latest revised manuscript and the highlighted version in Methods (line 919-920 in page 40).

C 2: the data demonstrates massive accumulation of the therapeutic compound in the liver, but there is not damage, at least by increase of enzymes that are released with liver damage. Have the authors looked at the liver for effects on other parameters as they did in the joint tissue? NF-kB, apoptosis etc? Assuming there is no effect on the liver, this should at least be discussed in the discussion.

R 2: We thank the reviewer for bringing up this important suggestion. We agree that the determination of NF- κ B signaling pathway and apoptosis in the liver is essential to elucidate whether the nanoparticles would induce any hepatic effect. Thus, we collected the liver tissues at the end of therapeutic process, followed by the immunochemistry analysis. As shown in **Supplementary Fig. 31a-e** and **Supplementary Fig. 32a-e**, the intraveous injection of FP/miR-23b nanoparticles did not trigger the alteration of TAB2, TAB3 and IKK- α expressions in comparison to the untreated mice, suggesting that the miR-23b delivery barely induced obvious activation of NF- κ B signaling pathway in the liver. Besides, no evident apoptosis was observed in liver tissues after the FP/miR-23b treatment. In addition, to figure out whether the accumulation of FP/miR-23b nanoparticles could alter the miR-23b expression in liver, we performed qPCR analysis for the miR-23b expression in liver tissues. As demonstrated in **Supplementary Fig. 31f** and **Supplementary Fig. 32f**, the hepatic miR-23b level was not significantly elevated in arthritic animals after several days post injection. Taken together, we sepeculated that the FP/miR-23b nanoparticles were gradually diminished by liver after the injection for several days, most of which were removed at the end of treatment, since hepatobiliary elimination has been widely reported for the nanoparticles with size of approximately 200 nm (Cornu, et al. *Toxicology* 2020, 430: 152344). Moreover, we searched the literatures, which suggested that the overexpression of miR-23b was able to provide the protective effect to the liver tissue by suppressing the inflammation and alleviating the fibrogenesis process (Bai, et al. *Front. Pharmacol.* 2020, 11: 173; Li, et al. *Exp. Cell Res.* 2021, 407: 112787.). Further, the fluorination of PAMAM could reduce the cytotoxicity of dendrimer due to the decreased positive charge after modification (**Supplementary Fig. 7**), which also contributed to the reduced hepatic damage. Collectively, we assumed that although FP/miR-23b nanoparticles accumulated in the liver tissue after the intravenous injection, nanoparticles were possibly scavenged by the liver, thus causing limited hepatic damage. Of note, more hepatic effects will be deeply interrogated in ongoing and future studies. As required, a detailed discussion has been added in the revised manuscript. Please refer

to Supplementary Fig. 31 and 32 and the highlighted version in Results (line 522-535 in page 23) and Discussion (line 618-624 in page 27).

REVIEWERS' COMMENTS

Reviewer #3 (Remarks to the Author):

The authors have addressed my comments.

Reviewer #4 (Remarks to the Author):

My concerns have been appropriately addressed in this revision.

Response to the comments from reviewers:

All authors would like to again express our gratitude to the reviewers for their valuable time and constructive comments.

Reviewer #3 (Remarks to the Author):

The authors have addressed my comments.

Response: We greatly thank the reviewer for all the comments that contributed to improve the quality of the manuscript.

Reviewer #4 (Remarks to the Author):

My concerns have been appropriately addressed in this revision.

Response: We greatly thank the reviewer for all the comments on this study.